# Vision-Language-Action Instruction Tuning: From Understanding to Manipulation

**Shuai Yang**[2,3*]   **Hao Li**[1,3*]   **Bin Wang**[2,3]   **Yilun Chen**[3‡]   **Yang Tian**[3]   **Tai Wang**[3]
**Hanqing Wang**[3]   **Feng Zhao**[1†]   **Yiyi Liao**[2†]   **Jiangmiao Pang**[3]
[1]University of Science and Technology of China, [2]Zhejiang University,
[3]Shanghai Artificial Intelligence Laboratory

## Abstract

To operate effectively in the real world, robots should integrate multimodal reasoning with precise action generation. However, existing vision-language-action (VLA) models often sacrifice one for the other, narrow their abilities to task-specific manipulation data, and suffer catastrophic forgetting of pre-trained vision-language capabilities. To bridge this gap, we introduce **InstructVLA**, an end-to-end VLA model that preserves the flexible reasoning of large vision-language models (VLMs) while delivering leading manipulation performance with the help of embodied reasoning. InstructVLA introduces a novel training paradigm, *Vision-Language-Action Instruction Tuning (VLA-IT)*, which employs multimodal training with mixture-of-experts adaptation to jointly optimize embodied reasoning and action generation on both standard VLM corpora and a curated 650K-sample VLA-IT dataset. On in-domain SimplerEnv tasks, InstructVLA achieves 33% improvement over SpatialVLA. To evaluate generalization, we introduce SimplerEnv-Instruct, an 80-task benchmark requiring closed-loop control and high-level instruction understanding, where it outperforms a fine-tuned OpenVLA by 96% and an action expert aided by GPT-4o by 29%. Additionally, InstructVLA surpasses baseline VLMs on multimodal tasks and exhibits inference-time scaling by leveraging textual reasoning to boost manipulation performance in both simulated and real-world settings. These results demonstrate InstructVLA's potential for bridging intuitive and steerable human-robot interaction with efficient policy learning. Project website.

## 1 Introduction

Large-scale pretraining has produced versatile foundation models in computer vision (CV) (Oquab et al., 2023; Radford et al., 2021) and natural language processing (NLP) (Bai et al., 2023; Touvron et al., 2023). Building on this progress, recent Vision-Language-Action (VLA) models (Black et al., 2024; Kim et al., 2024) adapt large vision-language models (VLMs) (Karamcheti et al., 2024; Beyer et al., 2024) and finetune them on embodied datasets to achieve generalizable manipulation. While the integration of multimodal reasoning has led to significant advances in VLMs (Wei et al., 2022; Liu et al., 2024a), such reasoning remains largely unexplored in VLA settings. Fully leveraging VLMs for reasoning-guided manipulation beyond VLA initialization remains an open challenge. Current attempts to incorporate the reasoning capabilities of VLMs into action learning face three main obstacles: (1) task interference, catastrophic forgetting (French, 1999) of multimodal ability during action training; (2) data scarcity, particularly the limited availability of manipulation datasets with rich multimodal supervision; and (3) methodological gaps, specifically the lack of effective mechanisms and training paradigm to translate multimodal reasoning into action generation. These limitations lead to a fundamental question for VLA-based manipulation:

*How can we acquire manipulation skills without eroding the VLM's multimodal reasoning, and how can such reasoning, in turn, enhance manipulation?*

To address these challenges and utilize VLMs more effectively, prior work has primarily adopted two strategies. The first aims to retain general multimodal capabilities while learning manipulation

---

*Equal contributions. ‡Project leader. †Corresponding authors: Feng Zhao, Yiyi Liao.

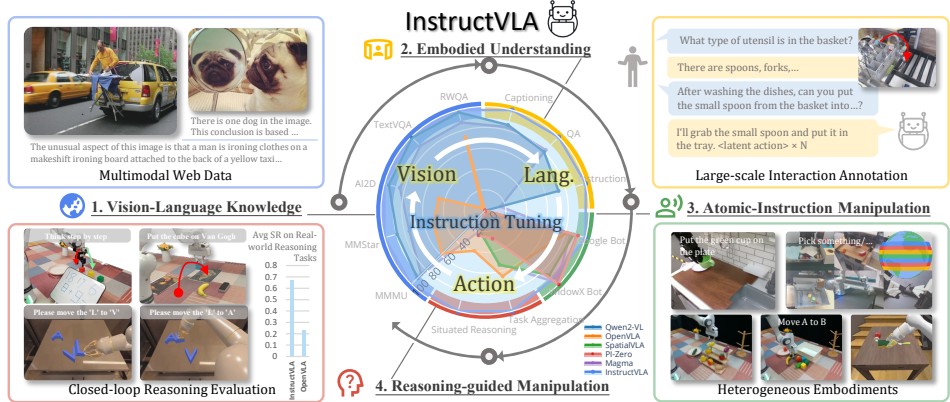

Figure 1: **Method overview.** InstructVLA integrates vision-language understanding with precise robotic control to achieve reasoning-guided manipulation. Its core training strategy, **Vision-Language-Action Instruction Tuning**, enhances manipulation by unifying general multimodal knowledge, embodied reasoning, and atomic instruction-based manipulation into a coherent chain of thought.

skills through unified auto-regressive modeling. Models such as RT-2 (Brohan et al., 2023) and Magma (Yang et al., 2025) follow this approach by co-training on vision-language and manipulation data. Yet, this paradigm often overlooks complex embodied reasoning, and our ablations reveal that the general VLM corpus exhibits a domain gap in embodied scenarios. The second strategy tightly integrates embodied reasoning into manipulation datasets to transfer VLM capabilities. Methods such as ECoT (Zawalski et al., 2024) and Emma-X (Sun et al., 2024) embed chain-of-thought (CoT) reasoning into manipulation datasets. While promising, these methods rely on action-pretrained architectures (Kim et al., 2024) and structured reasoning formats (e.g., subtasks, grounding), which limit expressiveness, suffer from catastrophic forgetting, and fail to demonstrate general multimodal capabilities-even with additional finetuning. Consequently, the extent to which VLM capabilities can be effectively translated into action generation in embodied contexts remains largely unexplored.

Building on these observations, we propose **InstructVLA**, a generalist VLA model that extends pretrained VLMs for accurate action generation while preserving strong multimodal understanding. Building on this unified modeling, we conduct extensive experiments to investigate how multimodal capabilities contribute to manipulation. Motivated by these insights, we design the **Vision-Language-Action Intruction Tuning** (VLA-IT) paradigm specifically tailored to bridge vision-language knowledge with action generation, treating language-conditioned action generation as an integral component of instruction following, as illustrated in Figure 1. To support this paradigm, we curate the **Vision-Language-Action Instruction Tuning dataset**, consisting of 650K human-robot interactions annotated with diverse instructions, scene captions, and question-answer pairs grounded in high-quality manipulation tasks (Ebert et al., 2021; Brohan et al., 2022). The training follows two stages: (1) *Action Pretraining*, which trains a VLM-driven action expert using latent action queries distilled from language-based motion descriptions, providing a learnable interface to the VLM while decoupling low-level control learning from the VLM backbone to preserve its multimodal reasoning capabilities; (2) *Vision-Language-Action Instruction Tuning*, which unifies language and latent action generation through a trainable mixture-of-experts(MoE) adaptation framework. This framework is jointly trained on multimodal datasets (He et al., 2024), manipulation datasets, and the curated VLA-IT corpus, enabling the automatic switch between textual reasoning and action generation, thereby effectively leveraging vision-language understanding and reasoning for action generation.

To validate the performance of InstructVLA, we introduce the **SimplerEnv-Instruct benchmark**, a manually designed evaluation suite featuring 80 zero-shot manipulation tasks. It encompasses both closed-loop manipulation tasks and high-level instruction reasoning, involving either situated understanding or decomposition into actionable subtasks. With its thinking ability during manipulation, InstructVLA outperforms the fine-tuned OpenVLA baseline by 96% and achieves a 29% improvement over an action expert model assisted by GPT-4o on SimplerEnv-Instruct, demonstrating its effectiveness in instruction following and task decomposition. Furthermore, InstructVLA surpasses similarly sized VLMs in multimodal performance and shows a 33% improvement over SpatialVLA in closed-loop manipulation (Li et al., 2024d). Our contributions can be summarized as follows:

- **Model.** We propose **InstructVLA**, a VLA architecture and training pipeline that **supports studying language capability in VLAs** by efficiently preserving pretrained vision-language knowledge from VLMs while integrating manipulation as a component of instruction following.

- **Dataset & Benchmark.** We design a **practical data and evaluation pipeline** for vision-language-action instruction following, supported by 650K tailored VLA-IT annotations and a manually curated benchmark suite, enabling evaluation of VLAs' instruction generalization capabilities.

- **Validation.** InstructVLA achieves leading performance across robotic manipulation tasks, multi-modal benchmarks, and real-world deployments, enabling intuitive and controllable manipulation.

## 2 RELATED WORKS

**Policy learning at scale.** Following the success of CV (Oquab et al., 2023; Zhai et al., 2023)and NLP (Touvron et al., 2023), recent research (Wang et al., 2024a; Brohan et al., 2022; 2023; Zheng et al., 2025; Wang et al., 2024b; Niu et al., 2025) shows that robot policies improve when trained in large heterogeneous datasets. RT-1 (Brohan et al., 2022) and RT-2 (Brohan et al., 2023), trained in large-scale real-world demonstrations, achieve strong in-domain accuracy and zero-shot transfer. Works such as Octo (Octo Model Team et al., 2024) and RT-X (Collaboration et al., 2023) extend this approach by aggregating the largest open-source manipulation datasets (Collaboration et al., 2023). Some methods, such as LAPA (Ye et al., 2024), Seer (Tian et al., 2024), and Moto (Chen et al., 2024b), use video generation and inverse dynamics to learn scalable motor representations. In the VLA domain, models are typically initialized from pretrained vision-language models (Kim et al., 2024; Qu et al., 2025; Brohan et al., 2023) leveraging prior visual-linguistic alignment instead of learning from scratch. Further, methods such as RT-Trajectory (Gu et al., 2023) and GraspVLA (Deng et al., 2025b) jointly train intermediate manipulation representations such as trajectories or bounding boxes using a combination of real and simulated data to guide action generation and enhance generalization.

**Vision-language-action models.** Recent foundation models (Brohan et al., 2023; Kim et al., 2024; Qu et al., 2025; Black et al., 2024; Chen et al., 2024b; Bjorck et al., 2025; Pertsch et al., 2025; Wu et al., 2026) integrate perception, language, and robot manipulation into a single network, using two main architectures. Autoregressive models such as RT-2 (Brohan et al., 2023), OpenVLA (Kim et al., 2024) and SpatialVLA (Qu et al., 2025) treat actions as discrete tokens. LLARVA (Niu et al., 2024) introduces 2D trace for pretraining. FAST tokenization (Pertsch et al., 2025) further compresses motion sequences. In contrast, flow-based VLAs avoid discretization; for example, $\pi_0$ (Black et al., 2024) and GR00T (Bjorck et al., 2025) generate actions through continuous flow matching (Lipman et al., 2022), while CogACT (Li et al., 2024a) and CronusVLA (Li et al., 2025a) use diffusion (Peebles & Xie, 2023). Hybrid approaches, like RoboDual (Bu et al., 2024), combine generalist action models with specialist action experts. Although flow-based methods (Black et al., 2024; Bjorck et al., 2025; Li et al., 2025a; 2024a) often achieve superior performance, they typically neglect the integration of autoregressive text reasoning (Brohan et al., 2023), which is crucial for leveraging the VLM's semantic capabilities. In contrast, our model unifies autoregressive VLM language generation with the flow-based action generation, demonstrating efficient co-training of language and action.

**Bringing step-by-step reasoning ability to manipulation.** Bridging pre-trained world knowledge to enhance the generalization of robot policies is a promising direction. One line of work standardizes intermediate representations (Li et al., 2026), such as primitive (Chen et al., 2024c), trajectories (Li et al., 2025b), keypoints (Li et al., 2024b) and masks (Huang et al., 2025). However, these approaches often rely on rule-based decomposition and hand-crafted planning heuristics, whose rigid separation from low-level control limits scalability and hinders end-to-end policy learning. CoT-VLA (Zhao et al., 2025) instead treats future video generation as an implicit chain-of-thought, but predicting image tokens step-by-step introduces computational overhead, limiting practicality for fast closed-loop control. More recently, unified modeling of perception, reasoning, and manipulation (Intelligence et al., 2025; AI, 2024; Shentu et al., 2024), along with other generative formulations (Pan et al., 2025; Zhou et al., 2024), has demonstrated the potential of leveraging pre-trained VLMs and LLMs for reasoning-guided generation, revealing emerging capabilities (Deng et al., 2025a). Yet, many prior studies depend on closed-source data (Intelligence et al., 2025) or conduct limited evaluation in real-world settings (Belkhale et al., 2024; Zhou et al., 2025), constraining reproducibility and large-scale assessment. Our work provides a pioneering exploration, supported by open data and benchmarks, to study *reasoning-guided manipulation* through the integration of reasoning and action.

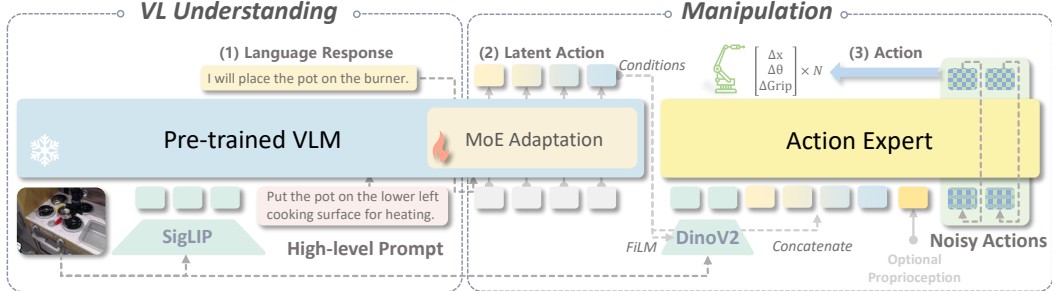

Figure 2: **Overview of the InstructVLA.** InstructVLA integrates the multimodal reasoning capabilities of a vision-language model with robotic manipulation. Generation consists of three steps: (1) asynchronous auto-regressive reasoning by the VLM, (2) latent action generation, and (3) action decoding. A MoE adaptation enables the VLM to alternate between reasoning and latent action prediction. The flow matching action expert decodes the final actions, conditioned on latent actions.

## 3 INSTRUCTVLA

We propose **InstructVLA** (Figure 2), a unified model for joint language-action generation that also mitigates task interference and catastrophic forgetting. Section 3.1 describes the architecture, including dynamic switching between reasoning and execution modes, as well as inference strategies. Section 3.2 presents the training paradigm for the instruction following of VLAs.

### 3.1 ARCHITECTURE

**Embodied VLM for text and latent action generation.** We propose a unified framework that enables simultaneous multimodal reasoning and language-steered latent action planning using a single VLM (Figure 2 (1) and (2)). The model produces textual outputs to preserve the strong language understanding and multimodal inference capabilities of the pretrained VLM, while subsequently generating latent action representations for downstream manipulation. To support action planning, we introduce $N$ learnable action queries $Q \in \mathbb{R}^{N \times D}$, which attend to the VLM's hidden states and extract task-relevant latent action $C \in \mathbb{R}^{N \times D}$, where $D$ is the VLM hidden dimension. Our implementation builds on the compact and efficient Eagle2-2B backbone (Li et al., 2025c), with a tailored training strategy described in Section 3.2. The VLM is supervised with cross-entropy on language output with loss $\mathcal{L}_{LM}$.

**MoE adaptation to harmonize reasoning and action.** A key challenge is enabling the model to seamlessly alternate between reasoning and manipulation. To this end, we adopt a MoE design (Zhou et al., 2022), which allows adaptive reweighting of expert modules based on input context and reasoning mode, thereby integrating multimodal reasoning with language-steered latent action. Specifically, LoRA (Hu et al., 2022) modules are employed as experts within the LLM backbone, preserving pretrained capabilities while ensuring efficient inference. A scalar head (Buehler & Buehler, 2024) predicts gating coefficients $\lambda_i$ for each expert by classifying the hidden state, enabling the model to adaptively blend their outputs. The resulting hidden states for $K$ experts are computed as $h = W_0 x + \sum_{i=0}^{K} B_i A_i x \cdot \alpha_i \cdot \lambda_i$, where $W_0$ is the original weight, $x$ denotes input, $A_i \in \mathbb{R}^{r \times d}$ and $B_i \in \mathbb{R}^{d \times r}$ are the LoRA parameters, $\alpha_i$ is the LoRA scalar factor, as detailed in Section F.2.

**Flow model as an efficient action expert.** To further decouple low-level control from high-level understanding, the action expert is designed to generate actions from image observations conditioned on VLM-derived intentions. It takes image features from DINOv2 (Oquab et al., 2023) vision encoder, latent actions, noisy action embeddings and optional information such as proprioception, and fuses these with a simple transformer architecture (Touvron et al., 2023) with block-wise causal attention. Specifically, non-causal attention is applied within each input, and causal attention between input types. The vision encoder, further enhanced with feature-wise linear modulation (FiLM) (Perez et al., 2018), plays a crucial role in directing actions to spatial and contextual input. The flow matching objective (Black et al., 2024) is used to supervise action learning, as detailed in Section F.3.

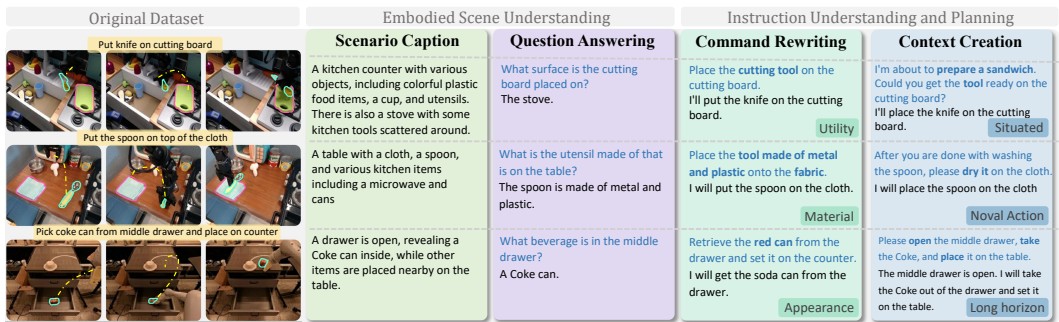

Figure 3: **Vision-language-action instruction tuning data examples.** Annotations focus on:
(1) improving scene understanding and (2) learning instruction following and planning.

**Inference.** InstructVLA integrates language and action generation in a single model with the following techniques to improve speed. (1) *Decoding strategies.* To mitigate the latency of autoregressive decoding, textual responses are generated via greedy search until the first action query token appears. The remaining action queries are then decoded in parallel within a single forward pass of the VLM. (2) *Language response and latent action caching.* We decouple language response from action generation by caching textual outputs across multiple action steps, leveraging their temporal stability. InstructVLA also supports cache latent actions, which reduces the number of VLM forward with minimal performance impact compared with ECoT (Zawalski et al., 2024) (see Section A.1).

## 3.2 TRAINING RECIPE

Direct co-training of vision, language, and action often leads to unstable optimization and slow convergence. We therefore adopt a principled two-stage training paradigm: first, action pretraining to align with the VLM's latent action embeddings; second, vision-language-action instruction tuning to integrate multimodal reasoning with manipulation.

**Stage 1: Action pre-training.** InstructVLA is pre-trained using heterogeneous manipulation data (Brohan et al., 2022; Ebert et al., 2021). To distill the knowledge from the VLM for manipulation, the model is trained to predict both actions and language motion (Section 4.1), with the latter supervised via cross-entropy loss. Due to the stability of flow matching and the next token prediction, the final loss is the direct sum of both losses as $\mathcal{L} = \mathcal{L}_{LM} + \mathcal{L}_{FM}$. During this stage, only the embedding of the latent action and action LoRA adapter on the LLM backbone are tuned, consisting of 650M parameters. The model trained is named the "Expert".

**Stage 2: Vision-language-action instruction tuning.** We extend visual instruction tuning (Liu et al., 2023) with a simple and efficient approach to train InstructVLA. Our key observation is that once the action expert has been pretrained to follow latent actions generated by the VLM, further adapting the LLM backbone enables the model to handle manipulation tasks with more complex instructions. In this stage, a language LoRA and a scalar head are added, which together with the stage 1 action LoRA constitute the MoE adaptation (Buehler & Buehler, 2024). This MoE module is the only trainable component in Stage 2, totaling 220M parameters. We detail the data pipeline for vision-language-action instruction tuning in Section 4.1; this data bridges pretrained vision-language capabilities with embodied task scenarios. To further bootstrap multimodal understanding, we co-train the model with additional multimodal datasets (He et al., 2024). The resulting model, referred to as the "Generalist", integrates both vision-language reasoning and manipulation capabilities.

## 4 VLA DATASET AND BENCHMARK

## 4.1 INSTRUCTVLA TUNING DATASET

We curate diverse hierarchical language annotations from large-scale manipulation datasets (Brohan et al., 2022; Ebert et al., 2021), including language motion (Belkhale et al., 2024) as detailed in Section D.1, along with the VLA-IT dataset for instruction tuning and reasoning transferring.

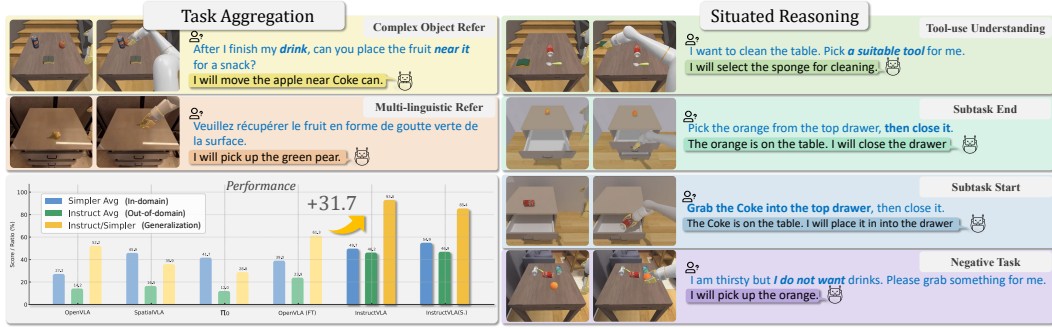

Figure 4: **Simpler-*Instruct*.** Six representative test cases with instructions and InstructVLA responses. Prior VLAs exhibit limited generalization compared to InstructVLA.

**Vision-language-action instruction tuning data.** To enable language-steerable VLA models, it is essential to curate diverse instructions, model responses, and reasoning patterns. We categorize our data into four types as illustrated in Figure 3. For embodied scene understanding: *(1) Scenario captioning* provides descriptions of the robot's environment *(2) Question answering* targets scene understanding through consistent QA pairs across an episode. Together, they bridge vision-language annotations with embodied scenes. For instruction understanding and latent action planning: *(3) Command rewriting* introduces instructional diversity through paraphrasing, attribute-based references and varied vocabulary. *(4) Context creation* generates implicit user goals or progress cues in multi-step tasks, requiring the robot to infer intent. These annotations support joint VLA reasoning.

We use GPT-4o (OpenAI, 2023) to annotate data with three frames from each episode, along with the corresponding instruction. Ground-truth instruction is crucial for annotation accuracy, emphasizing that even state-of-the-art VLMs can make errors in embodied tasks, leading to a performance gap when using GPT-4o as an instruction interpreter for such tasks. Additional details of the dataset analysis and prompt templates are provided in Section D.

## 4.2 SIMPLERENV-INSTRUCT

Building upon the SimplerEnv platform, we introduce **SimplerEnv-Instruct**, a benchmark specifically designed to evaluate the instruction-following and reasoning capabilities of vision-language-action (VLA) models in a zero-shot setting. Unlike prior manipulation benchmarks that primarily focus on atomic actions or low-level control, SimplerEnv-Instruct captures two essential yet underexplored abilities: (1) policy generalization to linguistic and visual diversity, and (2) contextual reasoning in situated environments, evaluated in the *situated reasoning* suite.

**Task creation.** We remove trivial cases and design novel tasks requiring genuine generalization rather than memorization. Novel objects and instructions are strictly out-of-distribution from the originals, and all tasks are cross-validated by three annotators for clarity and consistency. In total, we curated 80 tasks with 1.1K trials, about one third the size of SimplerEnv, keeping evaluation practical.

- **Task aggregation.** (50 tasks; examples shown in Figure 4, left). This suite assesses a model's ability to consistently interpret and execute core tasks based on both instructions and environmental context, despite variations in visual or linguistic forms. Tasks cover phenomena such as novel verbs, multilingual expressions, diverse object references, sentence rephrasings, and OOD objects.

- **Situated reasoning.** (30 tasks; examples shown in Figure 4, right). Beyond *task aggregation*, this suite evaluates a model's ability to reason over contextual cues or indirect instructions and to decompose commands into sub-goals. For example, "I want to clean the table. Pick a suitable tool for me." requires selecting the correct object (e.g., a sponge) from context.

Together, by leveraging the large-scale real-world training dataset, **SimplerEnv-Instruct** provides a reproducible benchmark that evaluates VLA generalization to unseen tasks. It achieves an affordable evaluation cost while systematically probing both task generalization and reasoning, filling a critical gap in VLA evaluation with a diagnostic, human-interpretable, and standardized benchmark.

Table 1: **Multimodal understanding.** #Params is the size of LLM backbone. S. denotes robot state.

| Methods | #Params | Multi-modal Understanding Benchmarks | | | | | | VQA Benchmarks | | | | | | |
|---|---|---|---|---|---|---|---|---|---|---|---|---|---|---|
| | | $MMMU^{Val}$ | MM-Vet | MMStar | $MME^P$ | OCRBench | HallB | MMB | TextVQA | DocVQA | InfoVQA | AI2D | ChartQA | RWQA |
| Bunny (He et al., 2024) | 8B | 43.4 | 39.1 | 45.4 | **1987.7** | 444 | 37.7 | 72.9 | - | - | - | 69.4 | 30.1 | 60.4 |
| PaliGemma (Beyer et al., 2024) | 2B | 34.9 | 33.1 | 48.3 | 1686.1 | 614 | 32.2 | 65.6 | 68.1 | 74.0 | 34.0 | 68.3 | 33.1 | 55.2 |
| Eagle2 (Li et al., 2025c) | 1.5B | 43.1 | **53.8** | **56.4** | 1572.1 | 818 | **45.8** | 74.9 | 79.1 | 88.0 | 65.8 | **79.3** | 82.3 | 63.1 |
| Qwen2-VL (Wang et al., 2024c) | 1.5B | 41.1 | 51.5 | 48.0 | 1872.0 | 809 | 41.7 | 74.9 | 74.9 | **88.6** | 61.4 | 74.7 | 73.5 | 62.9 |
| OpenVLA (Kim et al., 2024) | 7B | 0.0 | 0.0 | 0.0 | 0.0 | 0.0 | 0.0 | 0.0 | 0.0 | 0.0 | 0.0 | 0.0 | 0.0 | 0.0 |
| OpenVLA (FT) | 7B | 26.0 | 9.1 | 28.2 | 87.6 | 2.5 | 8.4 | 18.9 | 2.5 | 29.2 | 43.4 | 35.8 | 1.4 | 47.2 |
| ECoT (Zawalski et al., 2024) | 7B | 16.2 | 0.0 | 19.1 | 0.0 | 0.0 | 3.1 | 0.9 | 0.0 | 2.2 | 0.0 | 0.0 | 0.0 | 29.8 |
| ChatVLA Zhou et al. (2025) | 1.5B | 37.4 | - | 47.2 | 1435.2 | 729 | 39.9 | 69.0 | 71.2 | 83.3 | 53.3 | 67.6 | 59.9 | 57.0 |
| Magma (Yang et al., 2025) | 8B | 38.8 | 34.1 | 41.3 | 1496.5 | 518 | 38.0 | 69.7 | 66.5 | 65.4 | 45.2 | 66.1 | 61.8 | 56.5 |
| **InstructVLA-Generalist** | 1.5B | **44.2** | 51.7 | 56.2 | 1529.6 | 814 | 45.6 | 76.1 | 77.7 | 85.8 | 63.7 | 79.1 | 81.7 | 63.1 |
| **InstructVLA-Generalist(S.)** | 1.5B | 43.8 | 54.0 | 56.0 | 1548.0 | 829 | 42.8 | 76.3 | 78.2 | 86.0 | 63.7 | 78.9 | 82.9 | 63.5 |

Table 2: **Robotic manipulation.** Google and WidowX Robot denote two embodiments in SimplerEnv. For SimplerEnv-Instruct, we focus on two reasoning levels instead of embodiments. Magma[†] denotes evaluation with sampling. The results of InstructVLA are averaged over three random seeds.

| Methods | Google Robot | | | | | | | | WidowX Robot | | | Avg | SimplerEnv-Instruct | | |
|---|---|---|---|---|---|---|---|---|---|---|---|---|---|---|---|
| | Open/Close Drawer | | Put in Drawer | | Pick Coke Can | | Move Near | | Put Spoon | Put Carrot | Stack Blocks | | Task Aggregation | Situated Reasoning | Avg |
| | VM | VA | VM | VA | VM | VA | VM | VA | VM | | | | | | |
| RT-1-X (Collaboration et al., 2023) | 59.7 | 29.4 | 21.3 | 10.1 | 56.7 | 49.0 | 31.7 | 32.3 | 0.0 | 4.2 | 0.0 | 26.8 | - | - | - |
| RT-2-X (Collaboration et al., 2023) | 25.0 | 35.5 | 3.7 | 20.6 | 78.7 | 82.3 | 77.9 | 79.2 | - | - | - | - | - | - | - |
| RoboVLMs-2B (S.) (Li et al., 2024c) | 43.5 | 10.6 | 27.8 | 0.0 | 77.3 | 75.6 | 61.7 | 60.0 | 45.8 | 20.8 | 4.2 | 38.8 | - | - | - |
| OpenVLA-7B (Kim et al., 2024) | 63.0 | 28.8 | 0.0 | 0.0 | 18.0 | 60.8 | 56.3 | 67.7 | 4.2 | 0.0 | 0.0 | 27.2 | 14.8 | 13.6 | 14.2 |
| SpatialVLA-3B (Qu et al., 2025) | 57.4 | 41.8 | 0.9 | 9.1 | 86.0 | 88.0 | 77.9 | 72.7 | 16.7 | 25.0 | 29.2 | 45.9 | 23.6 | 9.8 | 16.5 |
| GR00T-N1.5-3B (S.) (Bjorck et al., 2025) | 27.8 | 13.2 | 7.4 | 2.2 | 51.7 | 63.6 | 51.0 | 54.0 | 62.5 | 45.8 | 16.7 | 36.0 | - | - | - |
| $\pi_0$-3B (S.) (Black et al., 2024) | 64.8 | 48.4 | 13.9 | 15.4 | 70.3 | 44.7 | 41.0 | 35.5 | 37.5 | 50.0 | 37.5 | 41.7 | 12.1 | 11.8 | 12.0 |
| **InstructVLA-Expert** | 52.3 | 61.7 | 50.3 | 33.1 | 79.6 | 92.3 | 68.3 | 71.9 | 43.1 | 40.4 | 9.7 | 50.9 | 21.6 ± 1.4 | 12.9 ± 0.4 | 17.3 |
| **InstructVLA-Expert(S.)** | 46.8 | 54.1 | 45.7 | 70.0 | 96.0 | 95.9 | 79.7 | 82.4 | 61.1 | 54.2 | 36.1 | 61.2 | 20.9 ± 0.3 | 20.5 ± 1.0 | 20.7 |
| Magma-8B (Yang et al., 2025) | 9.7 | 5.8 | 0.0 | 0.0 | 46.0 | 46.4 | 60.0 | 82.0 | 45.8 | 33.3 | 8.3 | 30.5 | 15.5 | 9.9 | 12.7 |
| Magma-8B[†] (Yang et al., 2025) | 56.0 | 53.4 | 6.4 | 18.5 | 83.7 | 68.8 | 65.4 | 65.7 | 35.5 | 31.0 | 12.7 | 43.6 | 26.2 | 21.4 | 23.8 |
| OpenVLA (FT) 7B | 63.9 | 42.6 | 3.7 | 6.9 | 62.3 | 88.7 | 65.8 | 67.7 | 12.5 | 33.3 | 4.2 | 39.0 | 28.3 | 19.5 | 23.9 |
| OpenVLA (FT&GPT) | - | - | - | - | - | - | - | - | - | - | - | - | 38.8 | 32.4 | 35.6 |
| **InstructVLA-Generalist** | 64.5 | 61.7 | 38.3 | 27.5 | 81.7 | 91.8 | 55.8 | 69.7 | 31.9 | 34.7 | 12.5 | 49.7 | 43.6 ± 1.4 | 48.8 ± 0.8 | 46.2 |
| **InstructVLA-Generalist(S.)** | 39.8 | 51.1 | 45.7 | 57.3 | 91.0 | 93.0 | 71.7 | 78.3 | 62.4 | 48.6 | 15.3 | 54.9 | 48.2 ± 1.3 | 45.6 ± 0.5 | 46.9 |

## 5 EXPERIMENT

**Benchmarks.** *(a) Multimodal:* We adopt automatic evaluation from VLMEvalKit (Duan et al., 2024), as detailed in Section E.1. *(b) SimplerEnv:* This benchmark (Li et al., 2024d) provides real-to-sim evaluation on large-scale manipulation datasets, incorporating visual matching and variance aggregation to assess generalization. *(c) SimplerEnv-Instruct:* As described in Section 4.2, this extension of SimplerEnv introduces novel objects, tasks, and instructions, offering a broader testbed for evaluating instruction generalization in VLAs. In addition, we assess embodied understanding in Section A.2 and manipulation performance on the LIBERO (Liu et al., 2024b) benchmark in Section A.3.

**Training details.** The VLM is trained with a resolution of $448 \times 448$ following Li et al. (2025c), while the action expert operates at $224 \times 224$ as in (Kim et al., 2024), using a fixed learning rate of 5e-5 without warm-up. The action expert employs a 12-layer transformer backbone with a hidden size of 768. Following Black et al. (2024), a $\beta$ distribution is used to enhance accuracy on the noisier time steps. During Stage 2 finetuning, manipulation and multimodal understanding are trained in an interleaved manner. Owing to our training paradigm, multimodal capabilities are preserved easily. We adopt a 1:7 multimodal-to-action training ratio, twice the ratio in ECoT and ChatVLA (1:3), reducing the additional computation needed to maintain multimodal ability. More details are in Section F.

**Baselines.** We categorize the baselines into three groups: (1) *Multimodal VLMs*, including Bunny(He et al., 2024), PaliGemma (Beyer et al., 2024), Eagle2 (Li et al., 2025c), and Qwen2-VL (Wang et al., 2024c); (2) *VLA models*, including RT-1-X and RT-2-X (Collaboration et al., 2023), RoboVLMs (Li et al., 2024c), SpatialVLA (Qu et al., 2025), $\pi_0$ (Black et al., 2024), GR00T-N1.5 (Bjorck et al., 2025), and OpenVLA (Kim et al., 2024); (3) *Generalist VLA models*, including Magma (Yang et al., 2025), OpenVLA fine-tuned (FT) from generalist pretrained model on both robotic and multimodal data, and ECoT(Bridge) (Zawalski et al., 2024). During evaluation, InstructVLA and other baselines use a temperature of 0 without sampling to expedite generation. We re-evaluate Magma with official checkpoint[1]. For ECoT, we report only its multimodal results due to its real-to-sim domain gap.

---

[1] We observe a notable performance gain for Magma when using sampling. Accordingly, we report its official score on SimplerEnv and re-evaluate its performance on SimplerEnv-Instruct under the sampling setting.

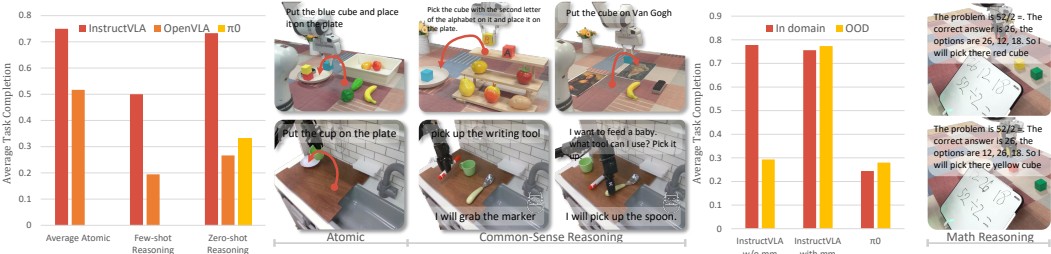

Figure 5: **Real-world experiments.** "Atomic" refers to atomic instructions. For the Kitchen and math settings, InstructVLA's responses are presented.

## 5.1 MAIN RESULTS

We present our main results in Tables 1 and 2. In Table 1, using the same generalist model InstructVLA (generalist), it not only outperforms the co-trained baseline Magma, but is also comparable to its base model Eagle2 and Bunny (VLM data corpus). InstructVLA further demonstrates stronger embodied understanding as detailed in Section A.2. In Table 2, InstructVLA (expert) outperforms the expert baseline SpatialVLA by 33.3% on SimplerEnv. Meanwhile, InstructVLA (generalist) not only maintains strong performance on SimplerEnv's atomic instructions but also achieves a 31.7% relative improvement on SimplerEnv-Instruct over the state-of-the-art baseline (OpenVLA with GPT-4o).

However, we observe that finetuning OpenVLA on multimodal and manipulation datasets does not fully restore its original multimodal capabilities, although it does improve task performance. Its performance can be further enhanced by integrating GPT-4o as an API-based system-2 module to rephrase instructions (OpenVLA (FT&GPT)). However, GPT-4o faces the same challenges in accurate instruction rewriting as noted in Section 4.1, and fails to outperform InstructVLA (Generalist). Methods such as Magma, which co-train both abilities of the VLM, better preserve multimodal ability, but still fail to match the performance of our approach. Although it also adapts two-stage training, ECoT relies solely on textual chain-of-thought reasoning over manipulation datasets and lacks the capability for multimodal question answering. We observe that it consistently generates manipulation-style CoT responses, without demonstrating effective instruction-following ability.

## 5.2 REAL-WORLD EXPERIMENTS

To evaluate InstructVLA in real-world scenarios, we conduct zero-shot experiments on the WidowX-250 Arm and few-shot experiments on the Franka Research 3 robot, as shown in Figure 5. The few-shot tasks involve spatial pick-and-place from a rack and cluttered tabletop setting and math-centric tasks detailed in Section A.5 to demonstrate the role of multimodal data. The zero-shot tasks are set in a kitchen environment following the Bridge dataset. InstructVLA is fine-tuned using the proposed training recipe, while OpenVLA is jointly trained on atomic skill and VLA-IT datasets with extra language supervision. The $\pi_0$ is finetuned using the official repository.

Each scenario includes both atomic and reasoning instructions. Atomic tasks emphasize in-domain objects and instructions with a focus on spatial generalization to assess baseline VLA capabilities. Both models perform comparably on direct in-domain instructions, but InstructVLA achieves a 23.3% improvement over OpenVLA. For reasoning tasks such as celebrity recognition, OCR, and tool-use inference, OpenVLA shows a substantial performance drop, whereas InstructVLA outperforms it by 41.7% in few-shot and 46.7% in zero-shot settings. On reasoning and math tasks, InstructVLA achieves a $2.5\times$ improvement over $\pi_0$, which behaves close to random guessing. Additional ablations and experimental setups are provided in Sections A.5 and H.

## 5.3 ABLATION STUDIES

We conduct ablation studies guided by two central questions: (1) Section 5.3.1. How can manipulation and multimodal understanding be effectively integrated into a single model through architectural design and training strategies? (2) Section 5.3.2. To what extent does vision-language comprehension

Table 3: **Ablation of action expert vision design and language motion.** "w/o Lang." denotes without using language motion. "w/o FiLM" denotes using only DINO. "w/o DINO" denotes action expert without the vision input.

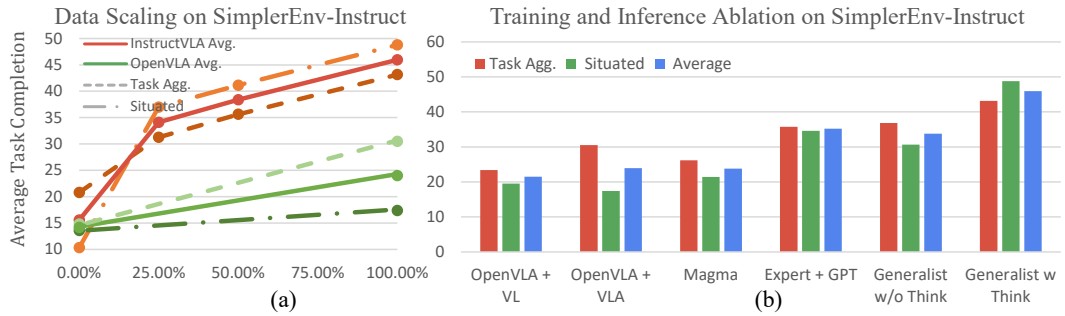

(a)                                          (b)

Figure 6: **Finetuning strategies.** (a) Freezing or finetuning the action head during VLA-IT training. (b) Training strategies when multimodal and manipulation tasks co-exist. "FFT" denotes full finetuning. "AR" denotes auto-regressive.

| Experts | WidowX Bot | Google Bot | Ave. |
|---|---|---|---|
| w/o DINO | 4.2 | 32.4 | 23.0 |
| w/o FiLM | 25.0 | 56.3 | 45.9 |
| w/o Lang. | 15.3 | 65.0 | 48.4 |
| InstructVLA | 29.1 | 64.8 | **52.9** |

Figure 7: **Data scaling and multimodal training.** Impact of scaling and training strategies on manipulation with multimodal reasoning.

enhance manipulation performance in complex scenarios? Through targeted ablations, we examine the impact of key architectural and training decisions on these capabilities.

### 5.3.1 MULTIMODAL AND MANIPULATION CO-TRAINING

**Strategies for multimodal and manipulation co-training.** As shown in Figure 6 (b), four paradigms are compared. (1) Following OpenVLA-OFT, FFT denotes full finetuning of the model with latent actions but without MoE adaptation and multi-stage training. With comparable computational resources, this setting yields suboptimal performance on both manipulation and understanding tasks. (2) The AR paradigm (Magma, RT-2) supports co-training but has limited performance. (3) Removing the MoE design while keeping the training paradigm preserves multimodal performance but reduces manipulation capability. (4) In contrast, InstructVLA leverages our proposed architecture and two-stage training strategy, achieving a 12.5% improvement over Magma on SimplerEnv.

**Language motion helps action understanding.** As shown in Table 3, introducing "language motion" (textual descriptions of low-level actions) supervision enhances the VLM's ability to associate visual cues with manipulation primitives, leading to a 9.3% improvement in overall success rate.

**Enhanced expert perception helps policy learning.** Incorporating richer perception into the action expert is efficient due to its compact design compared to the VLM backbone. As shown in Table 3, while the base VLM offers general visual understanding, fine-grained perception for manipulation tasks demands richer representations. Removing the DINOv2-based ViT encoder from the action expert results in a 50.0% performance drop, highlighting its critical role in capturing task-relevant visual cues. Incorporating FiLM to the ViT encoder yields a further 15.3% improvement by modulating visual features with latent actions. As shown in Table 2 the expert model with robot state generally performs better.

### 5.3.2 MULTIMODAL ABILITY TRANSFERS TO MANIPULATION

**VLM-only instruction tuning handles situdated understanding.** As shown in Figure 6(a), we examine the effect of VLA instruction tuning by comparing two configurations: (1) finetuning only the VLM, and (2) jointly finetuning both the VLM and the action expert. Freezing the action expert

achieves performance comparable to joint finetuning while substantially reducing the number of trainable parameters. This suggests that InstructVLA can effectively adapt complex and situated manipulation tasks by fine-tuning only the VLM, without altering the pretrained action expert.

**How VLA-IT scale and multimodal diversity affect reasoning-guided manipulation.** As shown in Figure 7(a), we first evaluate the **scaling behavior** of VLA-IT annotations on the SimplerEnv-Instruct benchmark. Situated reasoning tasks, which require grounding objects and goals in context, benefit most from larger annotation sets, highlighting the bootstrapped reasoning abilities inherited from VLMs. In contrast, pretrained OpenVLA fine-tuned on VLA-IT gains primarily from increased instruction diversity but shows limited improvement on situated reasoning tasks due to catastrophic forgetting of VL capabilities.

Table 4: **Effect of data dievrsity.** "T.A." denotes task aggregation, and "S.R." denotes situated reasoning on SimplerEnv-Instruct.

| QA & Cap. | T.A. | S.R. | Ave. |
|---|---|---|---|
| ✗ | 40.7 | 42.7 | 41.7 |
| ✓ | 43.6 | 48.8 | 46.2 |

These findings suggest that two-stage methods such as ECoT may be insufficient for fully leveraging the multimodal capacity of VLMs. We also examine the **effect of annotation diversity**, as shown in Table 4, where adding QA and captioning improves generalization of InstructVLA by 10.8%. Additional ablations are provided in Section A.4.

**Ablating strategies for incorporating reasoning into manipulation.** As shown in Figure 7 (b), (1) Simply combining manipulation and general multimodal ability through co-training does not yield significant benefits. Magma, despite co-training on multimodal datasets, shows limited transfer of vision-language capability to reasoning tasks on SimplerEnv-Instruct. Although OpenVLA suffers from catastrophic forgetting when finetuned with VLA-IT corpus, it still achieves better performance than Magma. (2) Multimodal ability can implicitly benefit manipulation when preserved through embodied reasoning annotation. Our generalist model, trained on the VLA-IT corpus, surpasses fine-tuned OpenVLA and Magma on the SimplerEnv-Instruct benchmark, even without explicit textual reasoning (*Think*). (3) Explicit textual reasoning further enhances manipulation. Enabling thinking in the generalist model brings a 36.1% performance gain over direct instruction execution and even outperforms InstructVLA-expert paired with GPT-4o as an external interpreter. Further analysis of the role of thinking is presented in Section A.1.

## 6 CONCLUSION

We present InstructVLA, a unified VLA model that integrates multimodal reasoning and action generation. We further demonstrate how the embodied understanding ability can directly benefit the maipulation tasks. Our data and training pipeline enables leading performance across manipulation tasks, multimodal benchmarks, and real-world deployments, paving the way for more generalizable, interpretable, and interactive robots.

## ACKNOWLEDGMENTS

This work is funded in part by the National Key R&D Program of China (2022ZD0160201), Shanghai Artificial Intelligence Laboratory, the Special Program of the Graduate School, University of Science and Technology of China, and Anhui Provincial Natural Science Foundation under Grant 2108085UD12. We acknowledge the research support from Shanghai Artificial Intelligence Laboratory, including computational and hardware resources. We also acknowledge the support of GPU cluster built by MCC Lab of Information Science and Technology Institution, USTC. The AI-driven experiments, simulations and model training were partly performed on the robotic AI-Scientist platform of Chinese Academy of Sciences.

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

# Appendix

## Contents

The supplementary material is organized as follows:

- Section A presents: (1) extended analysis of InstructVLA, (2) additional benchmarks on embodied understanding, (3) extra simulation benchmark and ablation study, (4) finetuning of OpenVLA under the same settings as InstructVLA, and (5) extra real-world ablation study.

- Section B discusses related concepts to InstructVLA and the proposed vision-language-action instruction tuning methods.

- Section C provides additional case analysis for InstructVLA, OpenVLA, and GPT-4o System2.

- Section D lists data annotation details, including GPT-4o prompt and dataset statistics. We further analyse the distribution of the instructions from two dimensions: task diversity and language diversity.

- Section E visualizes the SimplerEnv-Instruct benchmark and the acknowledgements of 3D assets.

- Section F details the model architecture, training configurations, inference speeds under different settings, and compute resources used.

- Section G shows several multimodal question answering examples.

- Section H describes the real-world experimental setup and provides example executions.

- Section I discusses the broader impacts, limitations, and outlines future directions for InstructVLA.

# A   MORE EXPERIMENTS AND ANALYSIS

## A.1   FURTHER DISCUSSIONS

Our further analysis is threefold. First, we present visualizations and scaling curves to examine the MoE and latent action designs. Second, we provide a detailed analysis of reasoning gains in manipulation tasks and case studies. Finally, we demonstrate that InstructVLA supports zero-shot dual-frequency generation to accelerate inference and compare the dataset scales used across different studies.

### A.1.1   EXTRA MODEL DESIGN ANALYSIS

The MoE and latent action are our key design components. We present an example illustrating the role of MoE under different task settings, including simple and reasoning instructions, with and without model reasoning. For latent action, we analyze its scaling behavior to guide future tuning.

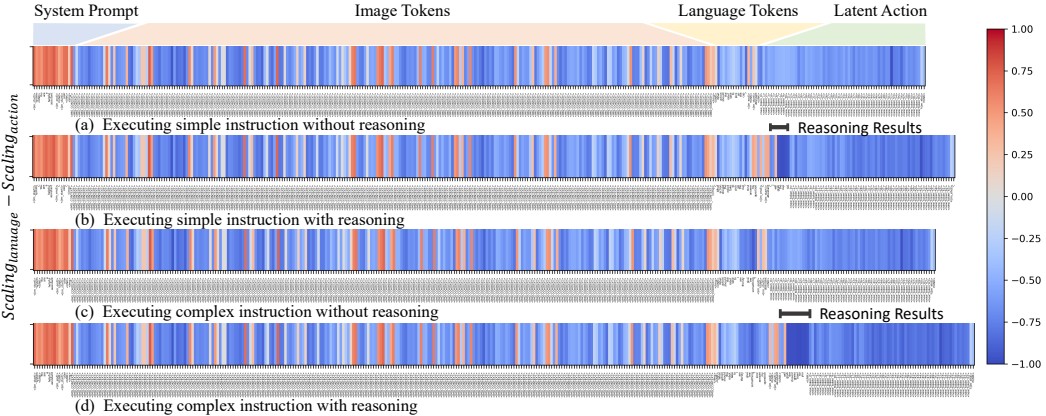

Figure 8: **Activation visualization.** We evaluate a WidowX zero-shot example across four settings. Red indicates stronger activation in the language adapter, while blue indicates stronger activation in the action adapter. The horizontal axis lists each language token. The generated tokens are marked.

**Analysis of MoE gating.** From the example in Figure 8, we draw the following intuitive conclusions:

- System prompts are primarily processed by the language adapter, reflecting its close connection to pretraining.
- Visual information is processed by both the language and action adapters, indicating that both semantic understanding and manipulation decision-making require visual inputs.
- During language generation, the model engages not only in multimodal reasoning but also in manipulation planning, as evidenced by the activation of the action expert. Notably, the action expert attends more strongly to nouns and verbs in the generated tokens, highlighting its role in instruction following.
- During latent action generation, the language expert plays a less prominent role. Instead, with multimodal reasoning, the model concentrates more effectively on action generation, as shown by the stronger activation of the action expert (deeper blue).

To conclude, the MoE has demonstrated its effectiveness in improving efficiency and handling heterogeneous datasets (Mu & Lin, 2025; Buehler & Buehler, 2024; Zhou et al., 2022; Team et al., 2025; Liu et al., 2024a). In InstructVLA, we further investigate how the MoE facilitates interleaved multimodal reasoning and manipulation decision making.

**Effects of latent action.** Latent action tokens are a key design component for decoupling high-level VLM planning from low-level action generation. As shown in Figure 9, we vary the number of tokens from 16 to 128. Too few tokens limit behavioral diversity, while too many reduce training efficiency. A setting of 64 offers a good trade-off under our current configuration.

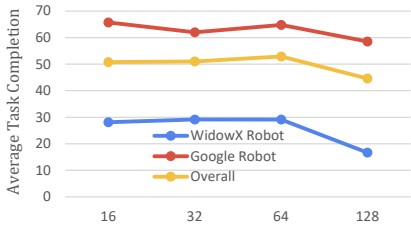

Figure 9: Impact of latent action token quantity on robot performance.

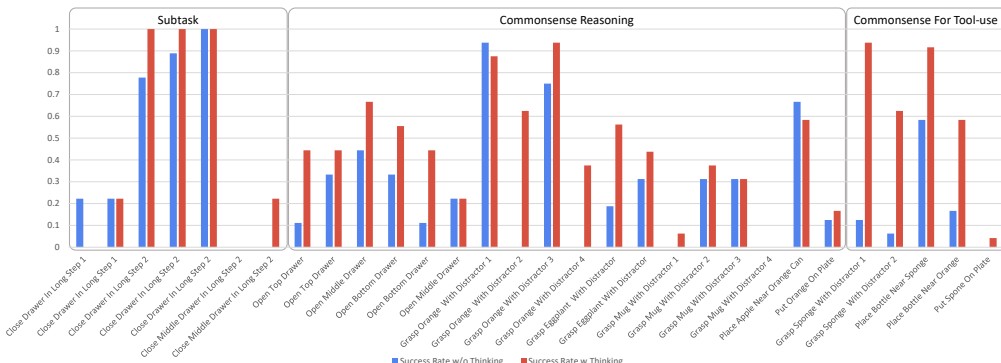

Figure 10: **Performance visualization** of 30 situated reasoning tasks with and without reasoning enabled. Activating reasoning in our generalist model generally improves performance. For clarity, tasks are grouped into three categories: *Subtask*, involving subtask identification; *Commonsense Reasoning*, requiring broad world knowledge; and *Commonsense for Tool Use*, focusing on tool-related reasoning.

### A.1.2    EXTRA REASONING-MANIPULATION ANALYSIS

In this section, we discuss the efficiency and design choices of VLA-IT training. We then analyze how multimodal reasoning benefits manipulation through fine-grained evaluation, examine its role in cross-embodiment generalization, and present a case study illustrating how a unique multimodal capability addresses challenging tasks.

**Effect of VLA-IT on Scaling and Reasoning.** As shown in Table 2, although the InstructVLA-expert model does not outperform the OpenVLA(OXE) on Situated Reasoning of SimplerEnv-Instruct, which benefits from direct full fine-tuning of the VLM backbone, InstructVLA-expert shows promising scaling ability in understanding complex instructions and performing test-time thinking after stage-2 VLA-IT training. This result reflects a deliberate design choice in InstructVLA, where latent action learning during pretraining focuses on querying from visual and simple instruction features rather than relying on the full semantic space of the VLM too early. This design offers two significant advantages. First, it preserves the original semantic space of the pretrained VLM, maintaining its vision-language capabilities. Second, it enables the model to integrate diverse reasoning contexts during VLA-IT training. These properties contribute to the strong performance gains achieved by our generalist model and demonstrate the effectiveness of this training paradigm.

**Embodied reasoning helps manipulation.** Allowing the model to perform test-time thinking by generating textual analysis of the given instruction can improve performance, particularly on situated reasoning tasks, as shown in Figure 11 (left). Notably, while the model with access to robot state outperforms the one without state when no instruction response is required, it provides limited performance gains when instruction following is involved. We hypothesize that state information helps the model retain manipulation skills but compromises its generalization to OOD environments and instructions.

**Fine-grained analysis of reasoning gains in manipulation tasks.** We compare the performance of the generalist model on SimplerEnv-Instruct with and without vision language reasoning, as

shown in Figure 10. A clear performance gap emerges in tasks involving commonsense tool use and interaction with articulated objects. This may result from instructions that do not explicitly state the intended actions and objects. For example, retrieving a cleaning tool from a drawer requires the robot to infer whether the prerequisite of an open drawer is satisfied, and to identify the sponge as the appropriate tool among several options. In addition to these cases, the reasoning process also improves performance on other situated reasoning tasks by grounding unfamiliar instructions using the pretrained in-domain knowledge of the vision language model.

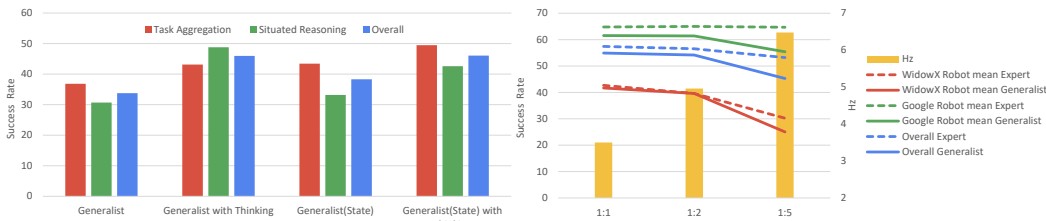

Figure 11: **Test-time tinking and dual-frequency evaluation.** "Expert" refers to the model after action pretraining, while "Generalist" denotes the model after VLA-IT tuning. For dual-frequency evaluation, the horizontal axis represents the ratio of VLM executions to expert model executions.

**VLA instruction tuning for cross-embodiment understanding.** To assess whether InstructVLA retains this capability, we evaluate three variants on SimplerEnv-Instruct (see Table 5): InstructVLA-Expert, trained solely on atomic instructions without test-time thinking; InstructVLA Generalist (Bridge), trained with the VLA-IT dataset on Bridge and the original Fractal dataset; and InstructVLA Generalist, trained with the full VLA-IT datasets across both environments. Adding the Bridge dataset results in a 139.4% improvement in Situated Reasoning performance for Generalist (Bridge) over the expert baseline, while task aggregation performance remains comparable. This discrepancy reflects differing generalization requirements: task aggregation emphasizes linguistic robustness, whereas Situated Reasoning demands vision-language grounding prior to action. The latter particularly benefits from the preserved reasoning capabilities of the pretrained VLM. As illustrated in Figure 12, the zero-shot model generates more diverse and accurate outputs than its fine-tuned counterpart.

Table 5: **Instruction tuning data ablation.** We evaluate three settings: without VLA-IT data, with data only on Bridge, and with VLA-IT data on both Fractal and Bridge. This ablation examines the contribution of the VLA-IT dataset and the cross-embodiment generalization of InstructVLA on SimplerEnv-Instruct.

| Instruction Tuning Data | | Name | Task Aggregation | Situated Reasoning | Overall |
|---|---|---|---|---|---|
| Bridge | Fractal | | | | |
| ✗ | ✗ | Expert | 20.8 | 10.4 | 15.6 |
| ✓ | ✗ | Generalist (Bridge) | 18.4 | 24.9 | 21.7 |
| ✓ | ✓ | Generalist | 43.3 | 48.8 | 46.0 |

**Case study on multimodal capability transfer.** As shown in Figure 13, we compare InstructVLA with OpenVLA (Kim et al., 2024), Magma (Yang et al., 2025), and CogACT (Li et al., 2024a), all using the same input (language instruction and a single image). **InstructVLA-Expert, though trained without multimodal and VLA-IT datasets, retains the OCR capability of the underlying VLM and achieves the best performance among baselines trained solely on manipulation data.** Finetuning InstructVLA-Expert into InstructVLA-Generalist with multimodal and VLA-IT datasets further enhances performance. For autoregressive models such as OpenVLA and Magma, multimodal finetuning improves OCR ability. In contrast, CogACT, when fine-tuned from OpenVLA(OXE) only on manipulation data with an action head, shows improved in-domain performance (on SimplerEnv) but suffers in generalization.

### A.1.3 EXTRA INFERENCE AND TRAINING ANALYSIS

**Dual frequency inference.** To further analyze the relationship between latent actions generated by the VLM and the final decoded actions, we decouple the inference frequencies of the VLM and the

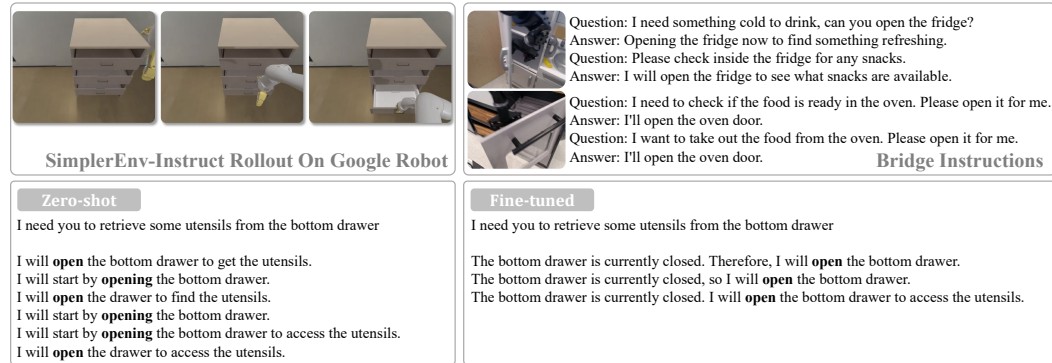

Figure 12: **Case study on cross-embodiment.** Top left: rollouts on SimplerEnv-Instruct. Top right: similar scenarios from the Bridge dataset with corresponding instructions. Bottom left: zero-shot results trained only on Bridge instructions. Bottom right: rollouts from the fine-tuned model.

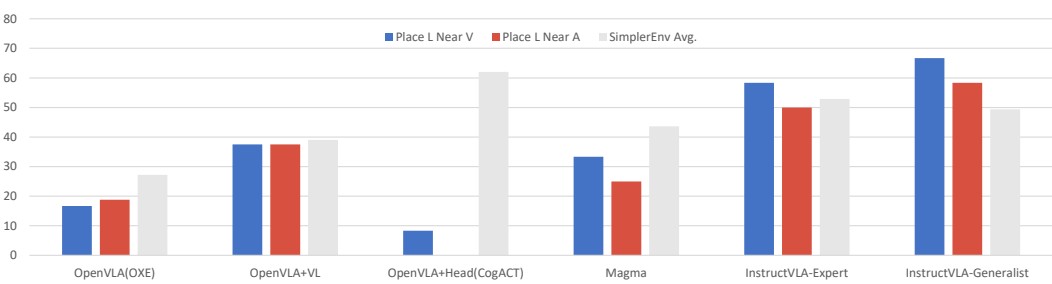

Figure 13: **Case study on multimodal capabilities.** OCR represents a unique multimodal skill of VLMs that is absent from typical manipulation datasets. We evaluate two tasks from the task aggregation set in SimplerEnv-Instruct, involving moving one letter to another (see Figure 18(1)). By comparing different finetuning paradigms, we assess how effectively multimodal capabilities are integrated into VLA models.

action expert, as illustrated in Figure 11 right. The results show that performance remains stable at a 1:2 ratio (VLM:expert), but begins to degrade at higher ratios. This suggests that latent actions offer relatively stable guidance to the action expert, reducing the need for frequent VLM queries.

**Training at scale.** A generalist VLA model with vision-language capabilities should be scalable across both manipulation and multimodal datasets. In this context, we compare datasets used by models claiming generalist abilities, as shown in Table 6. RoboMamba (Liu et al., 2024c) utilizes a limited manipulation dataset compared to other methods, while the dataset for ChatVLA (Zhou et al., 2025) is not reported. $\pi_{0.5}$ (Intelligence et al., 2025) employs a significantly larger multimodal dataset than other approaches, though its multimodal performance is not disclosed. Magma uses more robot and multimodal data but achieves slightly worse performance on both multimodal and manipulation benchmarks compared to InstructVLA.

Table 6: **Data comparison of different methods.** "Trans." denotes transitions.

|  | Magma(Yang et al., 2025) | ChatVLA(Zhou et al., 2025) | RoboMamba(Liu et al., 2024c) | $\pi_{0.5}$(Intelligence et al., 2025) | InstructVLA |
|---|---|---|---|---|---|
| Manipulation Data | 9.4M Trans. | - | 10K Trans. | >10000 Hours | 469 Hours/ 5.9M Trans. |
| Multimodal Data | 1.2M Images + 4M Videos | 54K | 1.5M | >7M | 2M |

## A.2 EMBODIED UNDERSTANDING EVALUATION

Table 7: **VLA-IT captioning evaluation.** "Sentence-BERT" and "SimCSE" represent learning-based evaluation methods, while the remaining metrics are traditional n-gram-based evaluations focused on word distribution.

| Methods | # Params | Sentence-BERT | SimCSE | BLEU-1 | BLEU-4 | METEOR | CIDER |
|---|---|---|---|---|---|---|---|
| Qwen2-VL (Wang et al., 2024c) | 1.5B | 61.3 | 67.5 | 16.8 | 1.5 | 12.4 | 0.30 |
| GPT4o (OpenAI, 2023) | - | 60.7 | 67.1 | 16.3 | 1.8 | 16.2 | 0.09 |
| OpenVLA(VLA-IT) (Kim et al., 2024) | 7B | 0.0 | 0.0 | 0.0 | 0.0 | 0.0 | 0.00 |
| Magma (Yang et al., 2025) | 8B | 59.8 | 66.7 | 12.4 | 1.2 | 12.3 | 0.12 |
| InstructVLA(Generalist) | 1.5B | 72.0 | 77.0 | 44.3 | 8.2 | 18.7 | 0.84 |

Table 8: **VLA-IT question-answering evaluation.**

| Methods | # Params | Sentence-BERT | SimCSE | BLEU-1 | BLEU-4 | METEOR | CIDER |
|---|---|---|---|---|---|---|---|
| Qwen2-VL (Wang et al., 2024c) | 1.5B | 51.9 | 53.4 | 15.3 | 2.8 | 17.9 | 0.82 |
| GPT4o (OpenAI, 2023) | - | 63.6 | 63.6 | 29.6 | 19.9 | 9.8 | 1.16 |
| OpenVLA(VLA-IT) (Kim et al., 2024) | 7B | 0.0 | 0.0 | 0.0 | 0.0 | 0.0 | 0.00 |
| Magma (Yang et al., 2025) | 8B | 53.5 | 54.5 | 23.7 | 5.7 | 21.6 | 1.04 |
| InstructVLA(Generalist) | 1.5B | 64.9 | 65.9 | 44.6 | 17.4 | 23.5 | 1.85 |

Table 9: **VLA-IT instruction response evaluation.** We use "context creation" annotations, as they present a more challenging and diverse set of instructions.

| Methods | # Params | Sentence-BERT | SimCSE | BLEU-1 | BLEU-4 | METEOR | CIDER |
|---|---|---|---|---|---|---|---|
| Qwen2-VL (Wang et al., 2024c) | 1.5B | 52.3 | 54.0 | 5.6 | 1.5 | 11.6 | 0.09 |
| GPT4o (OpenAI, 2023) | - | 52.8 | 54.1 | 17.8 | 4.2 | 20.6 | 1.02 |
| OpenVLA(VLA-IT) (Kim et al., 2024) | 7B | 0.0 | 0.0 | 0.0 | 0.0 | 0.0 | 0.00 |
| Magma (Yang et al., 2025) | 8B | 10.9 | 13.6 | 3.7 | 0.8 | 1.6 | 0.00 |
| InstructVLA(Generalist) | 1.5B | 71.6 | 73.1 | 50.2 | 24.1 | 25.8 | 2.26 |

In addition to the multimodal and closed-loop evaluations presented in the main results, we conduct supplementary language evaluations on the proposed VLA-IT dataset. This evaluation uses manually verified VLA-IT annotations on the Bridge dataset (Ebert et al., 2021), chosen for its diversity and distinct validation split. We generate 1,000 annotations following the method described in the VLA-IT dataset generation section. Two evaluation metrics are employed: (1) learning-based methods (Reimers, 2019; Gao et al., 2021), and (2) traditional metrics (Papineni et al., 2002; Young et al., 2023; Banerjee & Lavie, 2005).

The captioning, question-answering and instruction-following results are presented in Tables 7 to 9. We select Qwen2-VL (Wang et al., 2024c) and GPT-4o (OpenAI, 2023) as zero-shot VLM baselines, and include Magma (Yang et al., 2025) (zero-shot) and OpenVLA (Kim et al., 2024) fine-tuned on the VLA-IT dataset as baselines for VLA models.

Although OpenVLA is fine-tuned on the VLA-IT dataset, it fails to generate complete sentences under the same evaluation setting as InstructVLA, despite the performance on multiple-choice benchmarks reported in our main results. This suggests a significant loss of its free-form dialogue capability. Magma performs well on question answering and captioning tasks. However, it struggles with instruction response (Figure 14), often generating outputs misaligned with the given image. *We hypothesize that this failure stems from the similarity between these instructions and the atomic commands used in finetuning manipulation datasets, which disrupts the coherence of the language latent space near the action latent space.* This suggests a limited capacity to interpret and generalize free-form instructions, hindering effective transfer of vision-language capabilities.

InstructVLA achieves state-of-the-art performance, while GPT4o demonstrates competitive results. We visualize three episodes in Figure 15. GPT-4o generates more detailed captions but occasionally exhibits minor hallucinations. In the instruction response task, InstructVLA produces clearer and more grounded responses compared to GPT-4o, benefiting from the integration of ground-truth atomic instructions during the data annotation process, as discussed in Section D.4.

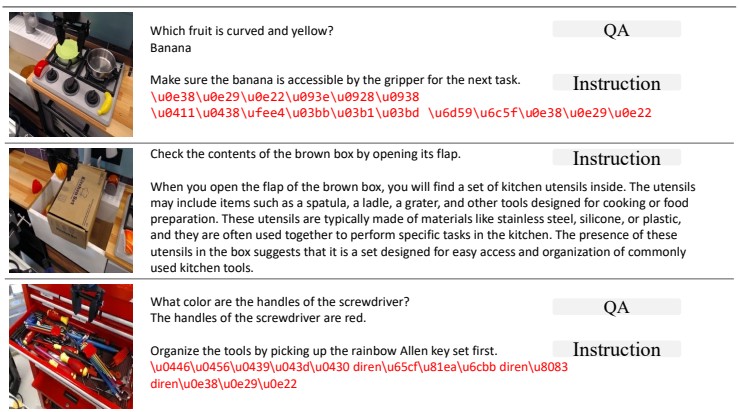

Figure 14: **Magma results.** Magma's responses collapse when given instructions resembling those in its manipulation tasks, possibly due to learned actions interfering with its language latent space.

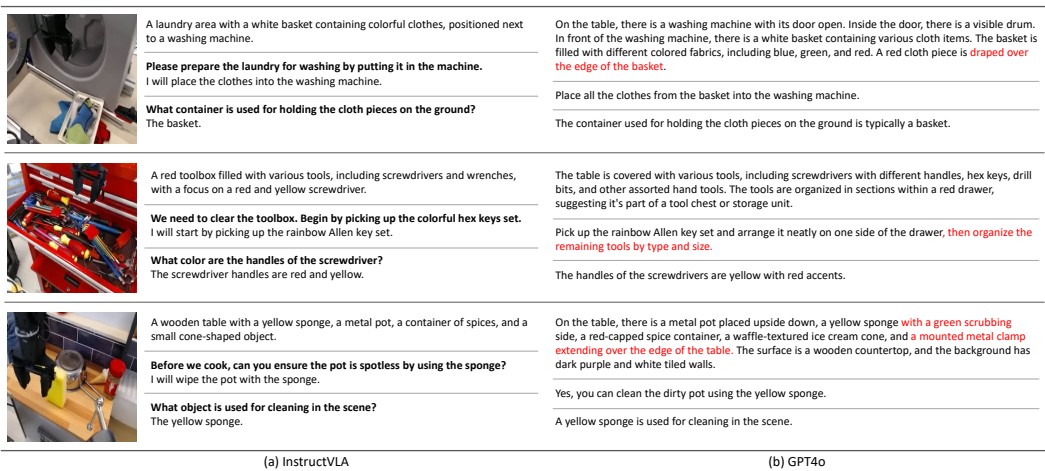

Figure 15: **Comparison with GPT-4o.** We visualize three examples from the VLA-IT language validation set. Each example includes a scenario caption (top), instruction response (middle), and question answering (bottom). The GPT-4o column displays **responses only**, as the instructions are identical across models.

## A.3 EXTRA MANIPULATION BENCHMARK

Table 10: **LIBERO benchmark results.** We present the success rate and standard error for each method across four task suites, which are averaged over three random seeds with 500 trials. "KI" denotes knowledge insulating from(Driess et al., 2025).

| | Spatial | Object | Goal | 10 (Long) | Average |
|---|---|---|---|---|---|
| OpenVLA-7B (Kim et al., 2024) | $84.7 \pm 0.9$ | $88.4 \pm 0.8$ | $79.2 \pm 1.0$ | $53.7 \pm 1.3$ | $76.5 \pm 0.6$ |
| OpenVLA-OFT-7B (Kim et al., 2025) | $97.6 \pm 0.9$ | $98.4 \pm 0.8$ | $97.9 \pm 1.0$ | $94.5 \pm 1.3$ | $97.1 \pm 0.6$ |
| SpatialVLA-2B (Qu et al., 2025) | $88.2 \pm 0.5$ | $89.9 \pm 0.7$ | $78.6 \pm 0.6$ | $55.5 \pm 1.0$ | $78.1 \pm 0.7$ |
| $\pi_0$-2B (Black et al., 2024) | $96.8 \pm 0.8$ | $98.8 \pm 0.9$ | $95.8 \pm 1.1$ | $85.2 \pm 1.2$ | $94.2 \pm 0.9$ |
| $\pi_0$-FAST-2B (Pertsch et al., 2025) | $96.4 \pm 0.7$ | $96.8 \pm 0.7$ | $88.6 \pm 1.0$ | $60.2 \pm 1.4$ | $85.5 \pm 1.0$ |
| CoT-VLA Zhao et al. (2025) | $87.5 \pm 1.4$ | $91.6 \pm 0.5$ | $87.6 \pm 0.6$ | $69.0 \pm 0.8$ | $81.1 \pm 0.6$ |
| GR00T-N1-1.34B (Bjorck et al., 2025) | $94.4 \pm 0.9$ | $97.6 \pm 1.0$ | $93.0 \pm 1.2$ | $90.6 \pm 1.0$ | $93.9 \pm 1.1$ |
| $\pi_{0.5}$ + KI (from scratch) (Intelligence et al., 2025) | 96.6 | 97.2 | 94.6 | 84.8 | 93.3 |
| $\pi_{0.5}$ + KI (from generalist model) (Intelligence et al., 2025) | 98.0 | 97.8 | 95.6 | 85.8 | 94.3 |
| InstructVLA (w/o wrist view) | 92.4 | 95.6 | 92.0 | 76.6 | 89.2 |
| InstructVLA-1.5B | $97.3 \pm 0.5$ | $99.6 \pm 0.0$ | $96.5 \pm 0.5$ | $89.8 \pm 1.6$ | $95.8 \pm 0.4$ |

**Benchmarks and baselines.** We evaluate InstructVLA on the LIBERO simulation benchmark (Liu et al., 2024b), which includes diverse robotic manipulation tasks in simulated environments. Following OpenVLA (Kim et al., 2024), we conduct experiments on four task suites, each containing 10 tasks with 50 human-teleoperated demonstrations. These suites assess spatial reasoning (LIBERO-Spatial), object type understanding (LIBERO-Object), task-oriented behaviors (LIBERO-Goal), and generalization to long-horizon tasks involving diverse objects, layouts, and goals (LIBERO-Long).

Our baselines fall into two categories: (i) generalist manipulation policies, including OpenVLA (Kim et al., 2024), OpenVLA-OFT (Kim et al., 2025), SpatialVLA (Qu et al., 2025), $\pi_0$(Black et al., 2024), and $\pi_0$-FAST(Pertsch et al., 2025); and (ii) manipulation policies with multimodal ability, including GR00T-N1 (Bjorck et al., 2025), and $\pi_{0.5}$(Intelligence et al., 2025) with knowledge insulation(Driess et al., 2025).

**Training details.** We augment InstructVLA with wrist-view images from the LIBERO training set (Liu et al., 2024b). Specifically, both the main and wrist-view images are provided to the VLM and the action expert. To reduce the tokenized input length, the two images are concatenated and resized into a single frame for VLM. Training follows the same hyperparameters as the Simpler-Env experiments and is performed on a single A800 node with 8 GPUs using a global batch size of 256, with evaluation every 1.5K steps.

**Results.** As shown in Table 10, InstructVLA achieves competitive performance despite not being pretrained on large-scale manipulation datasets like $\pi_{0.5}$(Intelligence et al., 2025; Driess et al., 2025) and using a much smaller VLM backbone than OpenVLA-OFT(Kim et al., 2025). Compared with recent VLAs such as $\pi_0$, InstructVLA attains higher performance with a substantially smaller action model (134M versus 300M).

## A.4 DATA ABLATION ON OPENVLA

Table 11: **Data ablation on OpenVLA.** "+VL" indicates finetuning OpenVLA with the same multimodal dataset used by InstructVLA. "+VLA-IT" refers to finetuning OpenVLA with the same VLA-IT dataset as InstructVLA. "+GPT4o" denotes using GPT4o as system 2 to translate free-form instructions into atomic ones.

| | OpenVLA (OXE) | OpenVLA + VL | OpenVLA + VL + VLA-IT | OpenVLA + VL + GPT4o | InstructVLA |
|---|---|---|---|---|---|
| Task Aggregation | 14.8 | 28.3 | 30.5 | 38.8 | 43.3 |
| Situated Reasoning | 13.6 | 19.5 | 17.4 | 32.4 | 48.8 |
| Average | 14.2 | 23.9 | 24.0 | 35.6 | 46.0 |

To investigate whether the performance gain of VLA-IT arises solely from the dataset itself, we reimplement the training procedure of the InstructVLA on OpenVLA (Kim et al., 2024), which represents a class of models trained under the action-only paradigm. As shown in Table 11, OpenVLA benefits from both vision-language and VLA instruction tuning data, with the latter showing greater

improvement in the task aggregation setting. This is attributed to exposure to more diverse instructions. However, performance on the situated reasoning setting remains unchanged, likely due to catastrophic forgetting caused by the action-only training paradigm, which limits OpenVLA's ability to leverage the VLM's reasoning ability through simple finetuning.

The greatest performance gain is observed when GPT-4o is introduced as an auxiliary System 2 in both evaluation settings. However, overall performance remains inferior to InstructVLA, as GPT-4o cannot fully ground free-form instructions to the atomic skills on which OpenVLA is pretrained.

## A.5 REAL-WORLD ABLATION

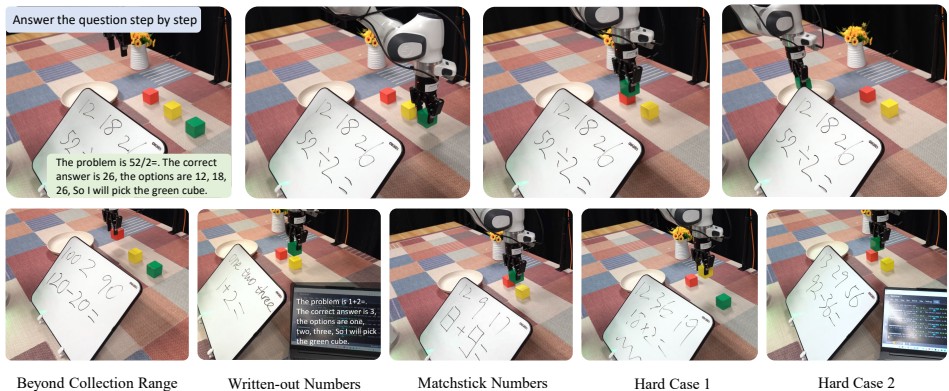

Figure 16: **Real-world ablation study.** The first row depicts the reasoning responses and the rolled-out actions, while the second row illustrates five categories of generalization.

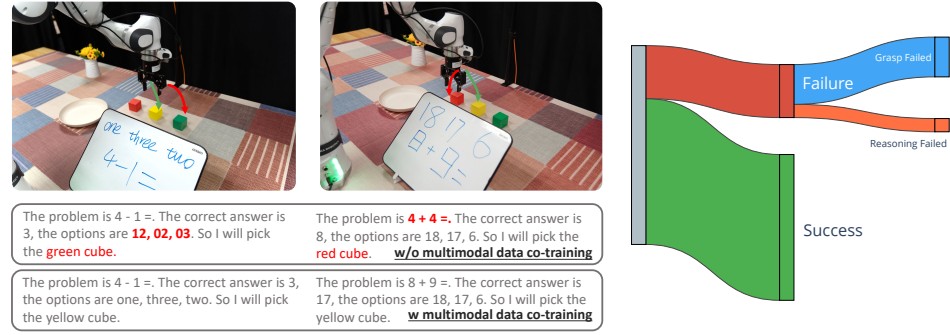

Figure 17: **Reasoning examples.** Two evaluation cases are presented to illustrate the role of multimodal datasets. We further summarize the results of InstructVLA in a Sankey diagram.

**Setup.** This case study evaluates the role of multimodal datasets in manipulation tasks. The robot setup follows our few-shot Frank evaluation. As shown in Figures 16 and 17, the model must first perform OCR to recognize the formula on the board and its answer options, then compute the result, and finally control the robot to grasp the correct object. This task mirrors a shopping scenario where robots often need to read prices and perform simple calculations to satisfy a requirement. The study jointly assesses OCR and calculation abilities, which are expected to benefit from multimodal data. To reduce bias, each case is evaluated three times with different target objects. In total, 250 training cases are collected but excluded from evaluation.

The in-domain tasks are defined as calculations within the range of the training data and written in a similar format. Generalization tasks are divided into five types: (1) Beyond Collection Range, (2) Written-out Numbers, (3) Matchstick Numbers, (4) Hard Case 1 (digits partially occluded with superimposed lines), and (5) Hard Case 2 (involving more complex calculations).

**Analysis.** By co-training with a general multimodal dataset, we observe that InstructVLA performs better on the tasks of *Written-out Numbers*, *Matchstick Numbers*, and *Hard Case 1*. We attribute this improvement to the inclusion of general OCR data within the multimodal dataset. Although the multimodal dataset is unfiltered (i.e., identical to the corpus used for training a VLM such as Bunny), it nonetheless enhances the instruction generalization for these specific tasks.

The SOTA VLA $\pi_0$ (Black et al., 2024), although pretrained on DROID Khazatsky et al. (2024), however, produces near-random results: although each grasp is executed precisely, the model frequently selects the wrong target object. Interestingly, when the third-view camera, which capturing the board with expressions and options, is masked, $\pi_0$ still behaves similarly. This suggests that $\pi_0$ largely ignores reasoning cues and overfits to the wrist view. While it performs precise grasping, the overall outcomes remain unsatisfactory.

## B  EXTRA RELATED WORKS

In this section, we delineate the distinctions between InstructVLA and several similarly named methods that differ substantially in their conceptual foundations and objectives.

### B.1  EMBODIED INSTRUCTION TUNING

**Vision-Action Instruction Tuning.** The concept of Vision-Action Instruction Tuning is introduced in LLARVA (Niu et al., 2024), which unifies robotic tasks through structured prompts and 2D trace supervision for cross-embodiment pretraining. In contrast, InstructVLA extends this idea by focusing on preserving the multimodal knowledge of VLMs and bridging high-level human instructions with low-level manipulation skills, enabling generalization to diverse tasks that require common-sense reasoning.

**Visuomotor Instruction Tuning.** The concept of Visuomotor Instruction Tuning is purposed in LLaRA (Li et al., 2024b). This approach formulates robot policies as visuo-textual conversations and produces 2D keypoints and rotations for manipulation. However, it functions primarily as a high-level planner, and its outputs require additional adaptation before being directly executed on robots.

### B.2  MULTI-STAGE TRAINING

**OpenVLA-OFT.** OpenVLA-OFT (Kim et al., 2025) extends OpenVLA (Kim et al., 2024) by incorporating FiLM layers, Parallel decoding, MLP action head, and has been applied to fine-tuning on smaller simulation datasets such as LIBERO Liu et al. (2024b). This approach demonstrates the effectiveness of architectural enhancements for improving manipulation performance in specific domains. However, while these techniques improve in-domain performance, they fall short in reasoning-centric settings such as SimplerEnv-Instruct, as shown in Figure 6 (b). In contrast, our work moves beyond architectural modifications by emphasizing generalizable manipulation with textual reasoning through MoE adaptation, latent action methods, and a comprehensive data and evaluation pipeline. With the proposed VLA-IT training paradigm, our generalist model achieves nearly a $2\times$ improvement over models that rely solely on architectural designs.

**Embodied Chain-of-Thought.** ECoT (Zawalski et al., 2024) introduces chain-of-thought (CoT) supervision to link reasoning with manipulation and follows a standard "pretrain-then-instruction-tune" paradigm. However, it relies on full-model pretraining fine-tuning, as in OpenVLA (Kim et al., 2024), which leads to catastrophic forgetting of vision-language capabilities. *In contrast, InstructVLA adopts a two-stage design: the first stage injects action-generation ability while deliberately preserving the multimodal knowledge of the pretrained VLM.* This approach ensures that the model retains open-world understanding and general multimodal reasoning, both of which are largely lost in ECoT. The second stage then strengthens multimodal reasoning and manipulation alignment. Consequently, InstructVLA supports broader inference modes (reasoning + manipulation, direct manipulation, and multimodal VQA) and achieves stronger performance with substantially fewer trainable parameters.

**Visual Chain-of-Thought.** CoT-VLA (Zhao et al., 2025) enhances manipulation by generating future image frames as visual chain-of-thought goals before predicting actions. While effective for goal specification, this approach relies on heavy video-generation supervision and does not exploit strong VLM pretraining for visual-language reasoning.

# C CASE STUDY

## C.1 REASONING CASES IN SIMPLERENV-INSTRUCT

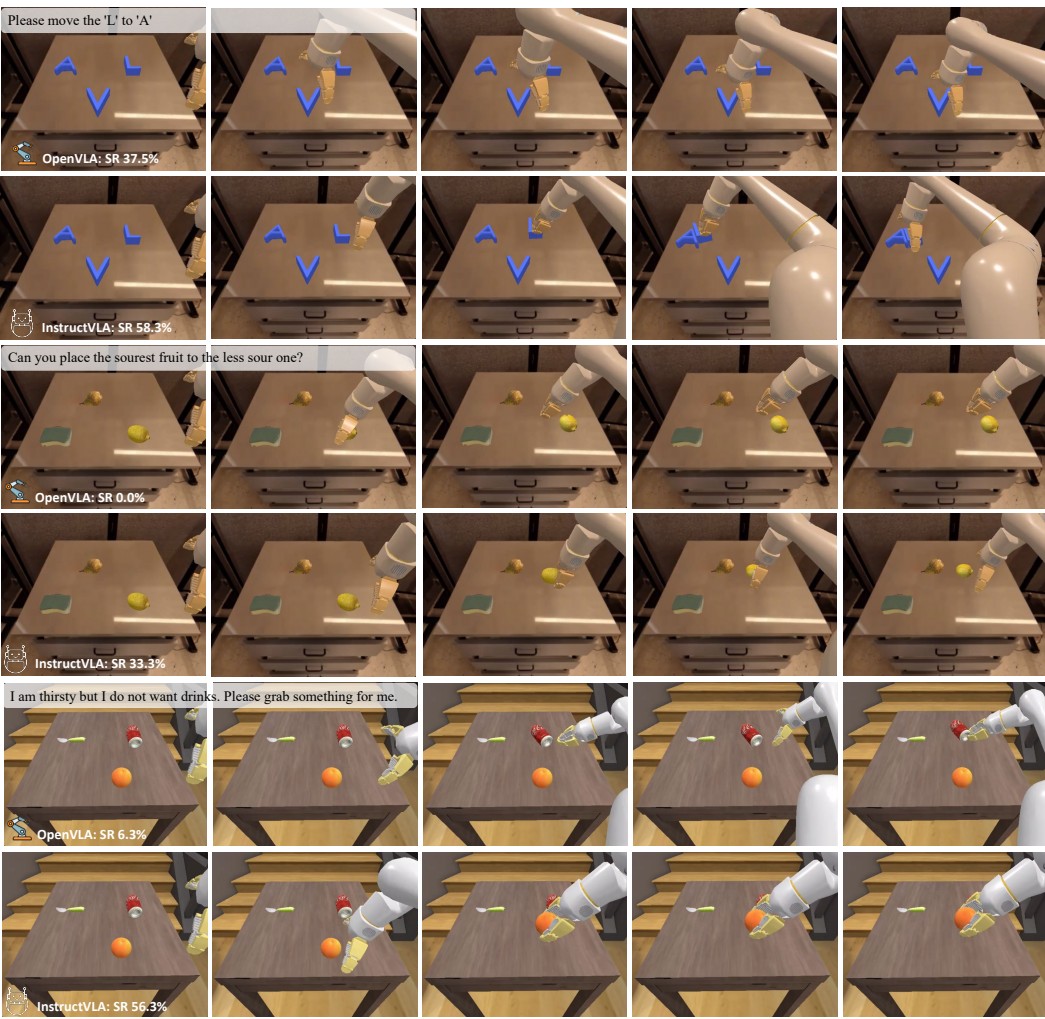

Figure 18: **Reasoning cases in SimplerEnv-Instruct.** Three cases of the VL fine-tuned OpenVLA and InstructVLA-Generalist. "SR" denotes success rate.

We present three representative reasoning cases in Figure 18. In the first example, OpenVLA fails to associate the letters "V" and "L" with their corresponding shapes in the image, resulting in consistent failure to grasp in all similar scenarios. In the second case, OpenVLA does not correctly associate the concept of "sour" with the corresponding fruit. As a result, its action is influenced by both the pear and lemon, leading to a grasp attempt between them that strikes the table. In the final example, OpenVLA fails to interpret the negation in the instruction and incorrectly grasps Coke instead of orange.

## C.2 FAILURE CASES

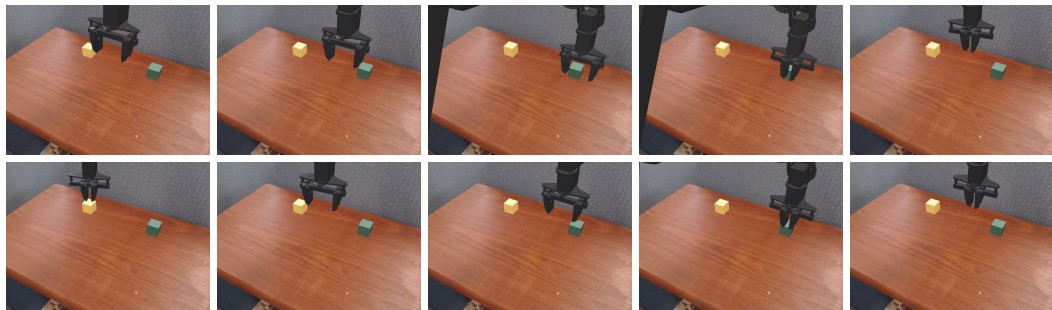

Figure 19: **Failure case 1 of InstructVLA.** The model receives only a third-person view image as visual input, making it difficult to estimate depth or the gripper's relative position to the object. Consequently, it fails to grasp the object accurately, despite the gripper appearing aligned with the target in the image.

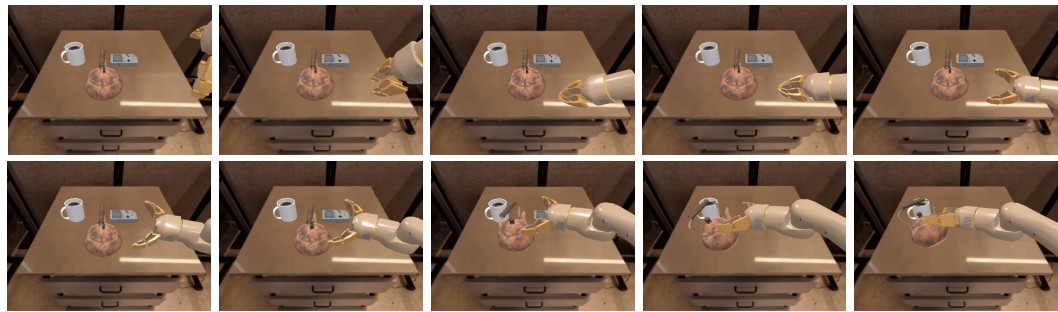

Figure 20: **Failure case 2 of InstructVLA.** The model fails to accurately estimate depth due to the real-to-sim gap, specifically the absence of arm reflection on the table, which causes the robot to become stuck in an out-of-distribution position.

We illustrate two representative failure cases of InstructVLA in Figures 19 and 20. While some failures may result from the real-to-sim gap, incorporating additional sensory inputs such as depth information and robot state may enhance performance. We leave this exploration for future work. Additionally, we observe that the model achieves higher success rates in language responses than in action execution, suggesting that multimodal understanding is more readily transferable than manipulation skills. This highlights a fundamental challenge in the development of embodied models.

## C.3 GPT4O AS THE AUXILIARY SYSTEM 2

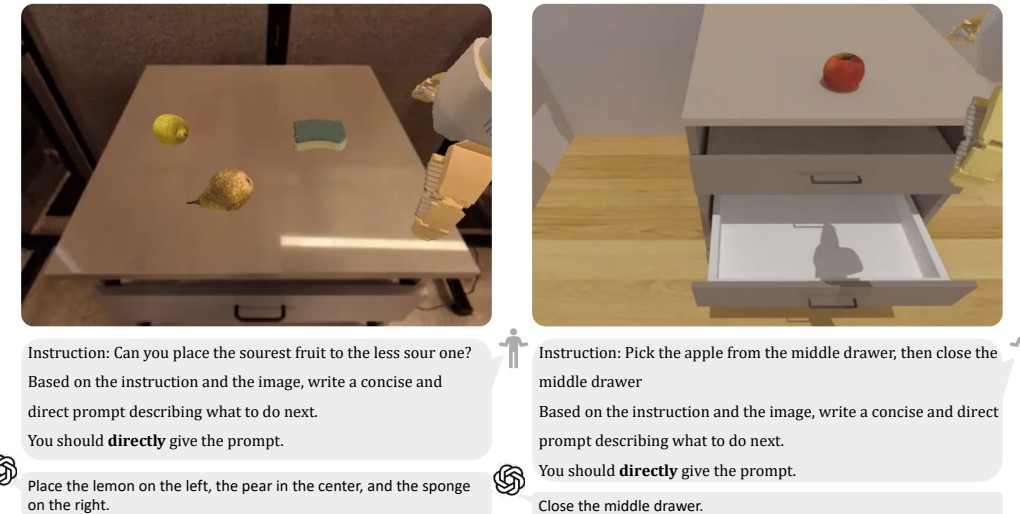

Figure 21: **GPT-4o as the auxiliary system 2.** We prompt GPT-4o with the first image from the environment along with the instruction, asking it to rewrite the prompt in a simple and clear format.

A strong baseline for InstructVLA integrates an expert model capable of executing atomic instructions with GPT-4o as an instruction parser to decompose complex, free-form commands for decision-making (Shi et al., 2025; Gao et al., 2025). The prompt used is listed in Prompt 1, and it was evaluated and refined on 20 test cases from the task aggregation to ensure reliable performance. Results on additional test cases are presented in Figure 21. GPT-4o successfully identified the atomic instruction in the second case but failed in the first.

During evaluation, GPT-4o is invoked only in the initial step to ensure an unobstructed view of the scene and to generate a free-form instruction. We do not provide a closed set of task-relevant instructions for selection, as the training set (Figure 23) lacks sufficient diversity in instructions and objects, and therefore does not adequately cover the evaluation settings. Across 80 evaluation cases, GPT-4o frequently fails in physical grounding, maintaining coherence, and accurately interpreting the scene.

---

**GPT-4o System-2 Prompt**

Instruction: Can you place the sourest fruit to the least sour one?
Based on the instruction and the image, write a concise and direct prompt to describe what to do next.
You should **directly** give the prompt.

---

Table 12: Definition of the 6D motion-primitive code $v \in \{-1, 0, 1\}^6$ and its natural-language verbalization. If no motion is detected, the default label is *stop*.

| Index | Symbol | Physical meaning | Values | Verbalization |
|---|---|---|---|---|
| 0 | $v_x$ | Translation along $x$ | $-1, 0, 1$ | backward / — / forward |
| 1 | $v_y$ | Translation along $y$ | $-1, 0, 1$ | right / — / left |
| 2 | $v_z$ | Translation along $z$ | $-1, 0, 1$ | down / — / up |
| 3 | $v_{\text{pitch}}$ | Pitch rotation | $-1, 0, 1$ | tilt down / — / tilt up |
| 4 | $v_{\text{yaw}}$ | Yaw rotation | $-1, 0, 1$ | rotate clockwise / — / rotate counterclockwise |
| 5 | $v_{\text{grip}}$ | Gripper motion | $-1, 0, 1$ | open gripper / — / close gripper |

# D   DATA ANNOTATION DETAILS AND ANALYSIS

The data analysis and GPT4o prompt are listed as follows (Figure 23 and Prompt 2).

## D.1   LANGUAGE MOTION PRE-TRAINING DATA

Language motion (Belkhale et al., 2024) provides intuitive linguistic descriptions of basic end-effector movements, which can be distilled into latent actions. For each episode, we extract a sequence of low-level motion primitives from the robot state trajectory. Let $s_t \in \mathbb{R}^8$ denote the state at time $t$, consisting of the end-effector position $p_t \in \mathbb{R}^3$, orientation quaternion $q_t \in \mathbb{R}^4$ (in $xyzw$ order), and the scalar gripper state $g_t \in \mathbb{R}$

$$s_t = (p_t, q_t, g_t). \tag{1}$$

We process overlapping windows of length $n$, $(s_t, \dots, s_{t+N})$, and summarize each window by the displacement between the first and last state

$$\Delta p = p_{t+N} - p_t, \quad \Delta q = q_{t+N} \otimes q_t^{-1}, \quad \Delta g = g_{t+N} - g_t. \tag{2}$$

The rotational displacement $\Delta q$ is converted to Euler angles $(r, \phi, \psi)$ in the xyz convention (roll, pitch, yaw). We then form a 6D continuous motion descriptor

$$d = (d_x, d_y, d_z, d_{\text{pitch}}, d_{\text{yaw}}, d_{\text{grip}}), \tag{3}$$

where $d_{x,y,z}$ correspond to the clipped $\Delta p$, and $d_{\text{grip}} = \Delta g$. Finally, we quantize each dimension with a symmetric threshold $\theta$:

$$v_i = \begin{cases} +1, & d_i > \theta, \\ 0, & |d_i| \leq \theta, \\ -1, & d_i < -\theta, \end{cases}$$

obtaining a discrete motion code $v \in \{-1, 0, 1\}^6$. This code is then mapped to a natural-language description (e.g., "move forward", "tilt up", "close gripper") using a fixed vocabulary, as shown in Table 12. If all dimensions are zero, the primitive is labeled as *stop*. We visualize example annotations from a representative episode in Figure 22. The language motions are concatenated into the model's response along with the corresponding user prompt.

## D.2   TASK DIVERSITY ANALYSIS

We categorize tasks into two broad classes: **Command Rewriting / Context Creation** and **Question Answering**. Each class includes several common task types:

### COMMAND REWRITING / CONTEXT CREATION

- **Complex Object Referencing:** Uses attributes, pronouns, or relational terms to reference an object.
  *Example:* "Place the red item next to the box."
- **Novel Action Referencing:** Rephrases a previously known action using a different verb or motion.
  *Example:* "Shut the drawer" (instead of "Close the drawer").

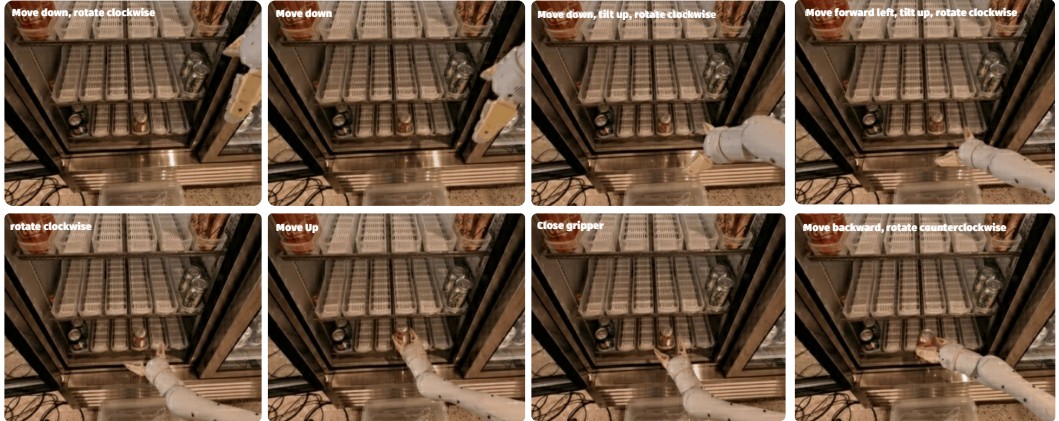

Figure 22: Language motion examples

- **Negative Task Specification:** Specifies the correct action by negating incorrect alternatives.
  *Example:* "I'm thirsty, but I don't want sparkling water—bring me something else."

- **Subtask Identification:** Isolates a step from a multi-step instruction with a clear sequential order.
  *Example:* From "Take the spoon out of the top drawer," execute only the first step.

- **Situated Task Identification:** Infers the required action based on contextual cues or situational conditions.
  *Example:* "I want to clean the table. What should I use?"

- **Direct Instruction:** Provides an explicit and unambiguous command.
  *Example:* "Organize the drinks by putting the green can next to the Coke can."

- **Tool-Use Understanding:** Refers to an object by its utility or function rather than its name.
  *Example:* "Hand me something to cut with" (instead of "Use the knife").

QUESTION ANSWERING

- **Quantitative Identification:** Requires determining the number or quantity of items.
  *Example:* "How many apples are on the table?"

- **Spatial Identification:** Involves spatial relationships between objects or with the user.
  *Example:* "Is the cup on the left or the right of the plate?"

- **Visual Identification:** Focuses on appearance-based attributes such as color or shape.
  *Example:* "Which one is the metallic silver object?"

- **Commonsense Answering:** Requires everyday reasoning or world knowledge.
  *Example:* "Which of these would you use to cut paper?"

- **State Identification:** Determines the current condition or status of an object.
  *Example:* "Is the drawer currently open or closed?"

The data examples for VIA-IT are provided in Figures 24 and 25.

### D.3 PROMPTING

The Prompt 2, along with three images captured at the beginning, middle, and end of each episode, is packaged and sent to GPT-4o. Episodes from the Bridge dataset (Ebert et al., 2021) that lack valid instructions are excluded from annotation.

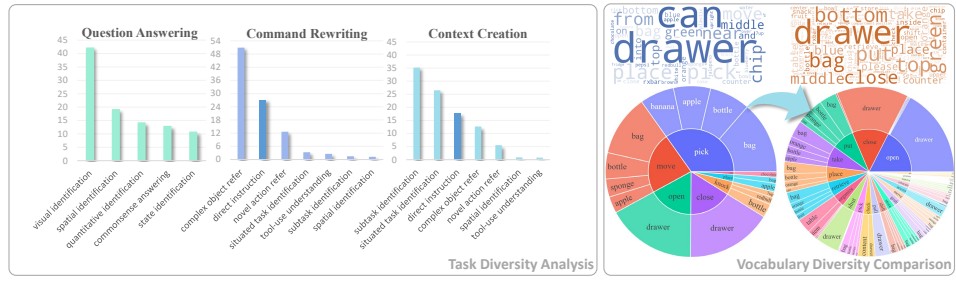

Figure 23: **Data analysis.** Left: We manually identify common task categories and calculate the distribution. The proportion of direct prompts is reduced in favor of more diverse, free-form expressions. Right: Word cloud and verb-noun analyses compare the original Fractal instructions with the VLA-IT corpus.

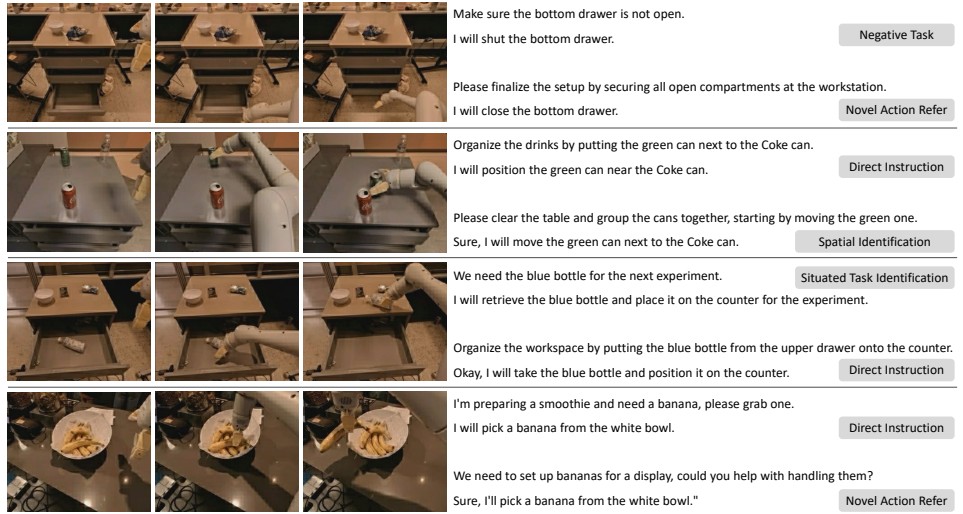

Figure 24: More VLA instructions on Fractal dataset.

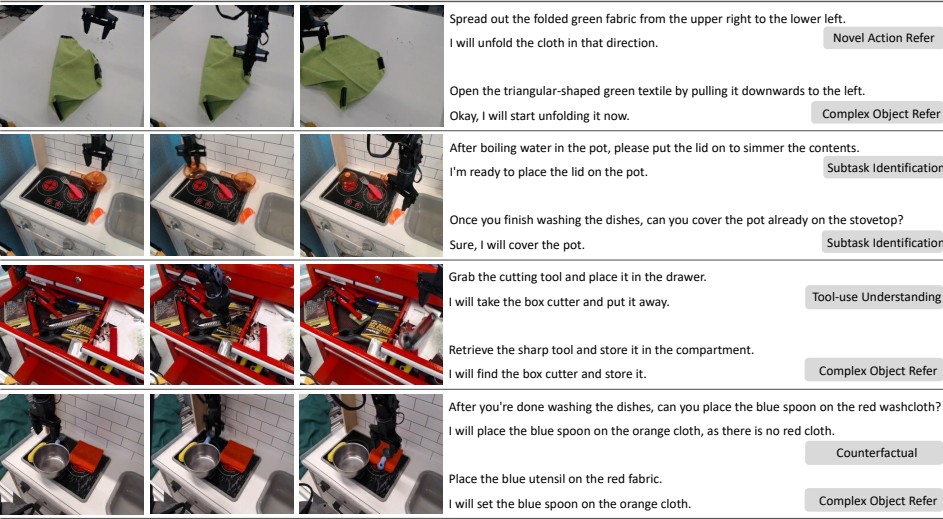

Figure 25: More VLA instructions on Bridge dataset.

---

**Data Annotation Prompt**

Imagine a robot assistant operating in a laboratory or household environment. The robot is expected to follow diverse commands based on realistic tasks and human interactions. Your task is to:

1. Write a caption to describe the visual scene shown in the **first image**. You should **NOT** include the robot itself here.

2. Based on the given robot task description and the images, generate new user instructions and corresponding robot responses with QA pairs.

The new user instructions should align with the actions performed by the robot in the images and with the environment shown in the images. You are required to produce three categories of instructions:

1. **Command Rewriting (CR)**: Rephrase the task description using diverse language styles and vocabulary. You may refer to objects by their utility, color, shape, or other attributes, but ensure the attribute you use is unique to each object.

2. **Context Creation (CC)**: Generate detailed scenarios where the robot needs to perform the given instruction. The situation should involve realistic surroundings or tasks where this instruction would be necessary. You may also simulate a long-horizon task based on the context provided by the image. Your generated question should **NOT** include the answer itself.

3. **Scene-related Commonsense QA (QA)**: Generate some other QA pairs that are related to the scene. The answer should be concise and consistent among the three images.

For each instruction, provide a concise robot response that clearly (use simple words) communicates the next action the robot will take. **Do not chain multiple actions together using phrases like "and then."** If necessary, the response may include a brief explanation of the reasoning. Avoid repeating the instruction in the response.

**Response Format**: You MUST respond in JSON format. You should include `"Caption"`, `"CR"`, `"CC"`, and `"QA"` in your response. You should create 1-3 entries for each of CR, CC, and QA.

**Example 1**: For the instruction "Close middle drawer":
*(Corresponding three images omitted)*
**Caption**: "A table with a Coke and chips on top, with its middle drawer open."

```
{
  "Caption": "A table with a Coke and chips on top, with its middle drawer open.",
  "CR": [ { "question": "Push the middle drawer closed.",
          "answer": "Ok, I will close it." },
        { "question": "Ensure the center drawer is closed.",
          "answer": "I will close the drawer." } ],
  "CC": [ { "question": "I want you to take out the Coke from the middle drawer and closing it.",
          "answer": "The Coke is on the table, and the middle drawer is empty. So, I should close the middle drawer." },
        { "question": "Please push the middle drawer shut so we can clear the workspace.",
          "answer": "Okay, I will close the middle drawer." } ],
  "QA": [ { "question": "What is in the middle drawer?",
          "answer": "The middle drawer is empty." },
        { "question": "How many Coke cans are on the table?",
          "answer": "One." } ]
}
```

**Example 2**: For the instruction "move the apple near the Coke":
*(Corresponding three images omitted)*
**Caption**: "A table with Coke, apple, and soap on it."

```
{
  "Caption": "A table with Coke, apple, and soap on it.",
  "CR": [ { "question": "Move the healthy food near the Coke.",
          "answer": "The healthy food refers to the apple, and I will move the apple to the Coke." },
        { "question": "Move the apple to the cylindrical–shaped object.",
          "answer": "Of course!" } ],
  "CC": [ { "question": "Gather all objects near the Coke, except the soap.",
          "answer": "I will move the apple to the Coke." } ],
  "QA": [ { "question": "I'm thirsty, what can I have?",
          "answer": "The Coke is on the table." },
        { "question": "What is the healthy food on the table?",
          "answer": "The apple." } ]
}
```

Your task description is "`<placeholder>`".
Now give your response in JSON format.

## D.4 GROUND TRUTH INSTRUCTION FOR DATA ANNOTATION

During data generation, we observe that GPT-4o often struggles to accurately interpret robot behavior using only the three provided images, performing noticeably worse than humans. To quantify this, we randomly sample 100 examples and prompt GPT-4o to generate our four types of annotations using a similar prompt (excluding the ground truth instruction from a human expert). We then manually evaluate the correctness of the results: a sample is scored as 1 if no obvious errors are found, 0.5 if minor errors are present, and 0 if completely incorrect.

The results are summarized in Tables 13 and 14, with two representative cases illustrated in Figures 26 and 27. In the first case, GPT-4o hallucinates the robotic arm as a bread roll, leading to an incorrect caption and instruction. In the second, it reverses the temporal order of actions, resulting in an inaccurate annotation.

We attribute this performance gap to GPT-4o's lack of temporal grounding and the low visual quality of images in manipulation datasets. In contrast, human-provided instructions inherently encode temporal links across the image sequence by grounding the task in context, identifying target objects, and specifying corresponding robot actions. This finding underscores that, despite their impressive capabilities, even state-of-the-art VLMs lack embodied experience and temporal grounding, limiting their ability to infer fine-grained actions in robot manipulation tasks.

Table 13: **Data annotation success rate.** GPT-4o shows a significant performance drop without ground truth instructions during data annotation.

| Method | Success Rate |
|---|---|
| With GT Instruction | 95.4% |
| Without GT Instruction | 45.0% |

Table 14: **Distribution of common error types.** Error analysis of GPT-4o annotations generated without access to ground truth instructions, with long-tail errors omitted.

| Error Type | Percentage |
|---|---|
| Ignore Vision Context | 32.5% |
| Reverse Temporal Order | 10.2% |
| Minor Object Hallucination | 5.7% |

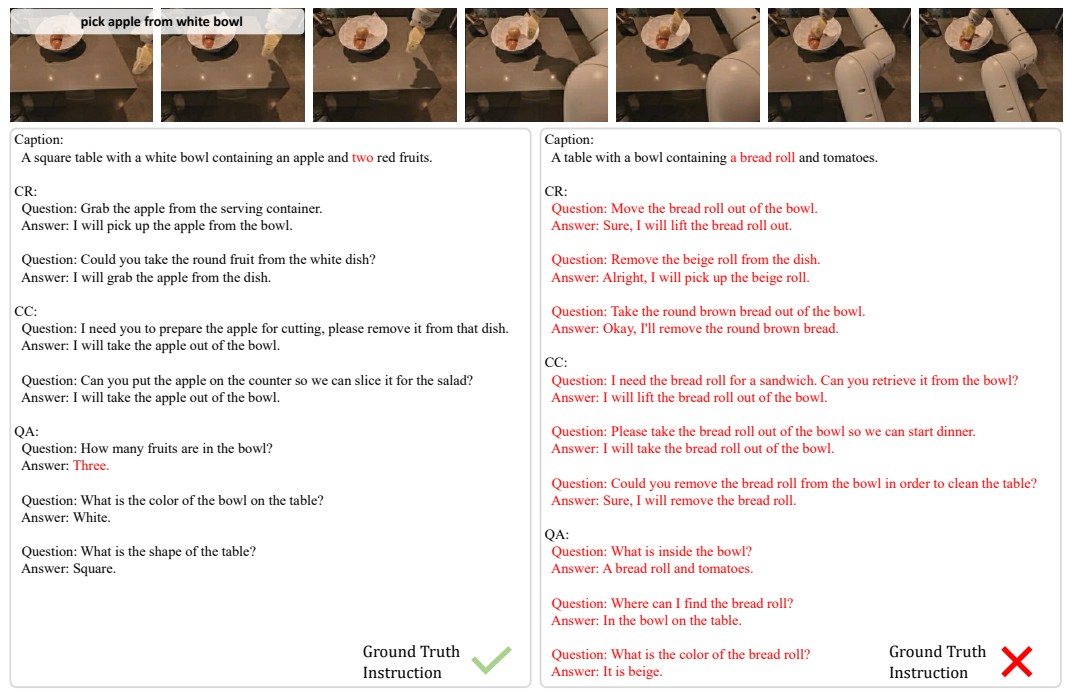

Caption:
A square table with a white bowl containing an apple and two red fruits.

CR:
Question: Grab the apple from the serving container.
Answer: I will pick up the apple from the bowl.

Question: Could you take the round fruit from the white dish?
Answer: I will grab the apple from the dish.

CC:
Question: I need you to prepare the apple for cutting, please remove it from that dish.
Answer: I will take the apple out of the bowl.

Question: Can you put the apple on the counter so we can slice it for the salad?
Answer: I will take the apple out of the bowl.

QA:
Question: How many fruits are in the bowl?
Answer: Three.

Question: What is the color of the bowl on the table?
Answer: White.

Question: What is the shape of the table?
Answer: Square.

Ground Truth Instruction ✓

Caption:
A table with a bowl containing a bread roll and tomatoes.

CR:
Question: Move the bread roll out of the bowl.
Answer: Sure, I will lift the bread roll out.

Question: Remove the beige roll from the dish.
Answer: Alright, I will pick up the beige roll.

Question: Take the round brown bread out of the bowl.
Answer: Okay, I'll remove the round brown bread.

CC:
Question: I need the bread roll for a sandwich. Can you retrieve it from the bowl?
Answer: I will lift the bread roll out of the bowl.

Question: Please take the bread roll out of the bowl so we can start dinner.
Answer: I will take the bread roll out of the bowl.

Question: Could you remove the bread roll from the bowl in order to clean the table?
Answer: Sure, I will remove the bread roll.

QA:
Question: What is inside the bowl?
Answer: A bread roll and tomatoes.

Question: Where can I find the bread roll?
Answer: In the bowl on the table.

Question: What is the color of the bread roll?
Answer: It is beige.

Ground Truth Instruction ✗

Figure 26: **Comparison of GPT annotations with and without ground truth instruction.** Errors are highlighted in red.

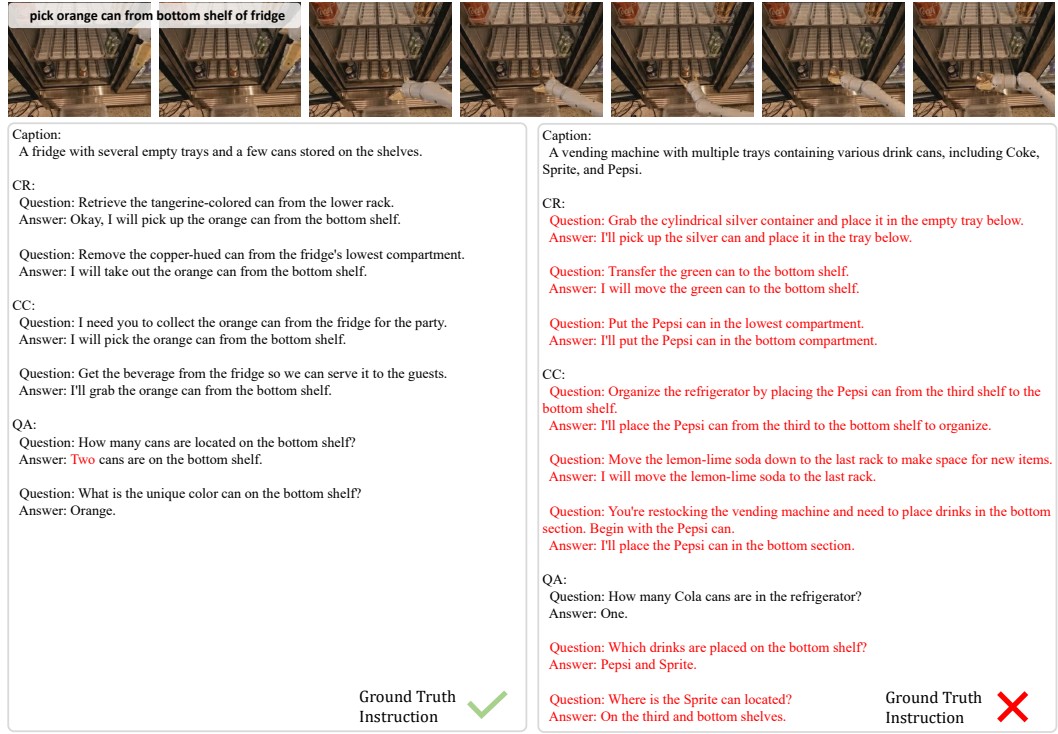

Caption:
A fridge with several empty trays and a few cans stored on the shelves.

CR:
Question: Retrieve the tangerine-colored can from the lower rack.
Answer: Okay, I will pick up the orange can from the bottom shelf.

Question: Remove the copper-hued can from the fridge's lowest compartment.
Answer: I will take out the orange can from the bottom shelf.

CC:
Question: I need you to collect the orange can from the fridge for the party.
Answer: I will pick the orange can from the bottom shelf.

Question: Get the beverage from the fridge so we can serve it to the guests.
Answer: I'll grab the orange can from the bottom shelf.

QA:
Question: How many cans are located on the bottom shelf?
Answer: Two cans are on the bottom shelf.

Question: What is the unique color can on the bottom shelf?
Answer: Orange.

Ground Truth Instruction ✓

Caption:
A vending machine with multiple trays containing various drink cans, including Coke, Sprite, and Pepsi.

CR:
Question: Grab the cylindrical silver container and place it in the empty tray below.
Answer: I'll pick up the silver can and place it in the tray below.

Question: Transfer the green can to the bottom shelf.
Answer: I will move the green can to the bottom shelf.

Question: Put the Pepsi can in the lowest compartment.
Answer: I'll put the Pepsi can in the bottom compartment.

CC:
Question: Organize the refrigerator by placing the Pepsi can from the third shelf to the bottom shelf.
Answer: I'll place the Pepsi can from the third to the bottom shelf to organize.

Question: Move the lemon-lime soda down to the last rack to make space for new items.
Answer: I will move the lemon-lime soda to the last rack.

Question: You're restocking the vending machine and need to place drinks in the bottom section. Begin with the Pepsi can.
Answer: I'll place the Pepsi can in the bottom section.

QA:
Question: How many Cola cans are in the refrigerator?
Answer: One.

Question: Which drinks are placed on the bottom shelf?
Answer: Pepsi and Sprite.

Question: Where is the Sprite can located?
Answer: On the third and bottom shelves.

Ground Truth Instruction ✗

Figure 27: **Comparison of GPT annotations with and without ground truth instruction.** Errors are highlighted in red. In this case, GPT-4o incorrectly infers the temporal sequence of actions without access to the instruction.

# E    BENCHMARK DETAILS

## E.1    MULTIMODAL

We use the automatic evaluation from VLMEvalKit (Duan et al., 2024) including MMMU(Val) (Yue et al., 2024), MMStar (Chen et al., 2024a), MME (Fu et al., 2024), OCRBench (Liu et al., 2024e), HallB(Avg) (Guan et al., 2024), MMB(Dev En V1.1) (Liu et al., 2024d), TextVQA (Singh et al., 2019), DoCVQA (Mathew et al., 2021), InfoVQA (Mathew et al., 2022), AI2D (Kembhavi et al., 2016), ChartQA (Masry et al., 2022) and RWQA (Team, 2024). These benchmarks collectively evaluate diverse multimodal capabilities, including general visual question answering, document, infographic and chart understanding, OCR reasoning, and hallucination robustness.

## E.2    SIMPLERENV-INSTRUCT

As shown in Table 15, although SimplerEnv-Instruct is primarily designed for instruction generalization, we incorporate diverse out-of-distribution objects, environments, and distractors to prevent VLA models from exploiting the benchmark by disregarding the instructions.

Table 15: Task distribution

| Attr. | with OOD Obj. | with OOD Env. | with Distract Obj. | Only Language OOD |
|---|---|---|---|---|
| Percentage(%) | 50.0 | 62.5 | 35.0 | 5.0 |

We select 10 task scenes with InstructVLA rollout actions and responses, as shown in Figures 28 and 29, to illustrate its performance on both task aggregation and Situated Reasoning tasks.

**Acknowledgements of 3D assets.** We gratefully acknowledge the creators of the following 3D assets used in *SimplerEnv-Instruct*. All assets are licensed under the Creative Commons Attribution license:

- *Fruit Bowl Collection*
  https://sketchfab.com/3d-models/fruit-bowl-collection-d89f6311cb0c4c37b0bf9cdd1e5abcb3
- *Coffee Mug*
  https://sketchfab.com/3d-models/coffee-mug-school-project-5f5ccee1514c440887c072fae8e0d699
- *Copper Tea Pot*
  https://sketchfab.com/3d-models/copper-tea-pot-27f2ac58f7614f2796630bdc6f18ee2f
- *Game Boy Classic*
  https://sketchfab.com/3d-models/game-boy-classic-0ae80019e6f046168923286d7e628f6f

All other assets are created using Blender or modified from SimplerEnv (Li et al., 2024d).

| Rollout Video | Instruction | Model Response |
|---|---|---|
| | Please retrieve the green teardrop-shaped fruit from the surface. Veuillez récupérer le fruit en forme de goutte verte de la surface. | I will pick up the pear. |
| | "Pick up the yellow fruit on the table saisis la banane | I will pick up the banana. |
| | Pick up the small round purple fruit | I will pick up the grape. |
| | Grab the green and white fizzy drink can from the workspace. toma la lata verde y blanca | I will pick up the 7up can. |
| | Please move the 'L' to 'V' | I will move the 'L' next to the 'V'. |
| | pick the plastic bottle and then place it near the orange. | I will pick up the bottle and move it to the orange. |
| | Can you place the sourest fruit to the less sour one? | I will put the lemon next to the pear. |
| | pick the mug and then place it near the play station. | I will pick the mug and move it closer to the play station. |
| | Pull the center drawer open | I will open the middle drawer now. |
| | Pull the top drawer out place the apple into the drawer you just opened | I will open the top drawer now. I will open the top drawer and place the apple inside. |

Figure 28: **Examples of task aggregation in SimplerEnv-Instruct.** We list ten examples with corresponding instructions and responses. Notably, InstructVLA shows the strong **zero-shot** ability to interpret multilingual instructions, recognize novel objects, and leverage OCR capabilities.

| Rollout Video | Instruction | Model Response |
|---|---|---|
|  | Please move the water bottle to wet the sponge. | I will pick up the water bottle to help you. |
|  | Please check if there are any items in the bottom drawer | The bottom drawer is currently closed, so I will open it to check for items.
The bottom drawer is already open, so I will check its contents. |
|  | I want a health vegetable. Grab it for me. | I will pick up the eggplant. |
|  | Can you grab the fruit before preparing the salad? | I will pick up the orange first. |
|  | I want to clean the table. Pick a suitable tool for me. | I will select the sponge for cleaning. |
|  | Open the top drawer and place coke can into it. | I will pick up the bottle and move it to the orange. |
|  | Pick the mug from the top drawer, then close the top drawer | The coffee mug is already on the table, so I will now close the top drawer. |
|  | put the tool that can be used to feed a baby on the plate | I will place the spoon on the plate. |
|  | After I finish my drink, can you place the fruit near it for a snack? | Sure, I will place the apple near the Pepsi can. |
|  | I am thirsty but I do not want drinks. Please grab something for me. | I will pick up the orange from the table. |

Figure 29: **Examples of Situated Reasoning in SimplerEnv-Instruct.** The second example's responses is recorded before and after the drawer is open.

# F  MODEL DESIGN AND TRAINING DETAILS

## F.1  INSTRUCTION FORMAT

To train captioning, question answering, and instruction-following capabilities, we integrate all tasks into a unified dialogue format. For captioning and question answering, we adopt the template shown in Prompt 3, where the captioning instruction is sampled from Prompt 4. For free-form instructions, we append the postfix "First answer my question." to elicit a direct response from the model, as illustrated in Prompt 5.

---

**Dialogue Format**

```
[
    {
        "role": "system", "content": DEFAULT_SYSTEM_MESSAGE
    },
    {
        "role": "user",
        "content": "[Question]",
        "image": image
    },
    {
        "role": "assistant",
        "content": "[Answer]"
    },
    {
        "role": "user",
        "content": "What action should the robot take to [Instruction]?"
    },
    {
        "role": "assistant",
        "content": "[Latent Action Queries]"
    }
]
```

---

**Caption Prompts**

- Describe what's on the table. Don't mention the robot arm.
- What objects are in the scene? Ignore the robot arm.
- Tell me what you see on the table, not the robot.
- Describe the items and their positions, but skip the robot.
- Look at the table and describe it. Don't include the arm.
- Only talk about the objects, not the machine.
- Give a short description of the scene, without the robot.
- Describe the setup on the table. Leave out the robotic arm.
- Focus on the objects and environment. Ignore the robot.
- Describe the environment and tabletop contents, excluding any robotic hardware.

---

**Instruction Format**

```
[
    {
        "role": "system", "content": DEFAULT_SYSTEM_MESSAGE
    },
    {
        "role": "user",
        "content": "What action should the robot take to [Instruction]? First answer my question.",
        "image": image
    },
    {
        "role": "assistant",
        "content": "[Response] [Latent Action Queries]"
    }
]
```

---

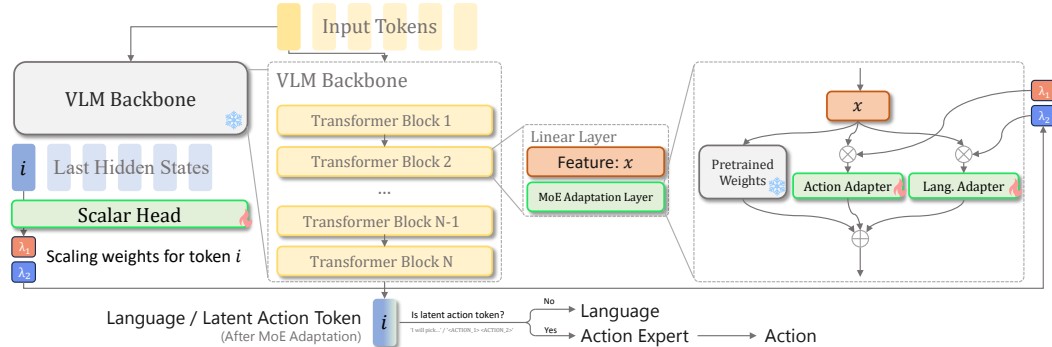

Figure 30: **Detailed overview of the MoE adaptation architecture.** The frozen VLM backbone's last hidden states are classified by a scalar head to produce gating weights $\lambda_1$ and $\lambda_2$, which control the weighted MoE adaptation. Similar to finetuning VLMs with multiple LoRA adapters, the MoE adaptation computes a weighted sum over the LoRA experts. The predicted tokens are then used differently based on their token type: language tokens are directly decoded as the model's response, while features corresponding to action tokens are decoded by the action expert (see Figure 2 (right)) to produce continuous actions.

### F.2 MoE Adaptation

We briefly outline the LoRA (Hu et al., 2022) mechanism, which serves as the basis for MoE adaptation design Buehler & Buehler (2024). LoRA models weight updates as lying in a low-dimensional subspace by freezing the original weights $W_0 \in \mathbb{R}^{d \times k}$ and parameterizing updates through a low-rank decomposition:

$$W_0 + \Delta W_0 = W_0 + BA, \tag{4}$$

where $B \in \mathbb{R}^{d \times r}$ and $A \in \mathbb{R}^{r \times k}$ with $r \ll \min(d, k)$. The forward pass becomes

$$h = W_0 x + \Delta W_0 x = W_0 x + BAx. \tag{5}$$

In practice, LoRA introduces a fixed scaling factor $\alpha$, yielding

$$h = W_0 x + \alpha BAx. \tag{6}$$

As shown in Figure 30, the scalar head in MoE adaptation predicts a gating coefficient $\lambda$ to reweight two LoRA adapters, inspired by mixture-of-experts architectures (Zhou et al., 2022). This is achieved by rescaling each adapter's scaling factor :

$$\alpha_i^* = \alpha_i \cdot \lambda_i, \tag{7}$$

where $i$ indexes the LoRA adapters. The scalar head together with the LoRA adapters constitutes the MoE adaptation module.

Unlike X-LoRA (Buehler & Buehler, 2024), which trains the scalar head and LoRA adapters separately. We first pretrain the action adapter in Stage-1 following the standard LoRA pipeline. In Stage-2, we introduce the language adapter for embodied reasoning and the scalar head, and train the complete MoE adaptation module jointly. We instantiate the scalar head as a 4-layer MLP with dimensions shown in Table 17 for simplicity, without any other auxiliary loss design.

### F.3 Learning Objective and Inference Procedure

We adopt flow matching (Black et al., 2024; Lipman et al., 2022) to learn the action chunk $\mathbf{A} \in \mathbb{R}^{H \times 7}$ (Zhao et al., 2023) over a horizon $H$. The training objective is defined as the flow matching loss:

$$\mathcal{L}_{FM} = \mathbb{E}\left[\|V\theta(\mathbf{A}^\tau, q_t) - (\epsilon - \mathbf{A})\|^2\right], \tag{8}$$

Table 16: Overview of data used in Stage-1 action pretraining and Stage-2 VLA instruction tuning.

| Supervision Type | Self-Annotated | Stage-1 | Stage-2 |
|---|---|---|---|
| Action | ✗ | ✓ | ✓ |
| Language Motion | ✓ | ✓ | ✓ |
| General Multimodal Datasets | ✗ | ✗ | ✓ |
| Embodied Reasoning(VLA-IT dataset) | ✓ | ✗ | ✓ |

Table 17: **Model parameters.** "Adapter" and "Scalar Head" are used for MoE adaptation. Specifically, two LoRA adapters are used to learn latent action generation and assistant response during VLA-IT.

| Component | Parameter | Value |
|---|---|---|
| Adapter | Rank | 128 |
| | Alpha | 256 |
| | Dropout | 0.05 |
| | Target | Attn. Q/K/V/O |
| | | MLP Up/Down |
| Scalar Head | Hidden Size | $2048 \rightarrow 128 \rightarrow 128 \rightarrow 128 \rightarrow 2$ |
| | Activation | ReLU |
| Action Backbone | Depth | 12 |
| | Head | 12 |
| | Hidden Size | 768 |
| | RoPE Theta | 1000 |
| Proprioception Encoder(Optional) | Hidden Size | $8 \rightarrow 768 \rightarrow 768$ |
| | Activation | SiLU |
| Action Encoder with Time Embedding | Hidden Size | $7+768 \rightarrow 1536 \rightarrow 768$ |
| | Activation | SiLU |

where $\tau \in [0, 1)$ denotes the flow step, and $V_\theta(\mathbf{A}^\tau, q_t)$ is the network output conditioned on $q_t$, which encodes information from DINOv2 (Oquab et al., 2023) and a latent action $C$. The interpolated noisy action is given by $\mathbf{A}^\tau = \tau\mathbf{A} + (1 - \tau)\epsilon$, with $\epsilon \sim \mathcal{N}(\mathbf{0}, \mathbf{I})$.

During inference, we generate the action chunk using forward Euler integration:

$$\mathbf{A}^{\tau+1/N} = \mathbf{A}^\tau + \frac{1}{N}V_\theta(\mathbf{A}^\tau, q_t), \qquad (9)$$

starting from $\mathbf{A}^0 \sim \mathcal{N}(\mathbf{0}, \mathbf{I})$, with $N = 10$ denoising steps. For language prediction, we use the standard cross-entropy loss. We simply sum the two losses with a 1:1 weighting. The data we used are detailed in Table 16.

## F.4 MODEL PARAMETERS

Additional model parameters are provided in Table 17, with flow-matching sampling settings detailed in Table 18. All projectors—including those aligning latent actions and DINO-ViT visual features to

Table 18: **Flow matching parameters.** The time steps is sampled from $p(\tau) = \beta(\frac{s-\tau}{s}; 1.5, 1)$ (Black et al., 2024)

| Component | Parameter | Value |
|---|---|---|
| Flow Sampling | s | 0.999 |
| | Inference Steps | 10 |
| Sinusoidal Time Embed | Max Period | 100 |

the action expert's dimension—use a simple two-layer MLP with SiLU activation. The action head, also a shallow MLP with SiLU, maps the action expert's hidden states to $\mathbb{R}^{N \times 7}$, where $N = 16$ is the prediction horizon and 7 denotes the action dimension, including the gripper.

## F.5 INFERENCE SPEED

We evaluate the inference speed of InstructVLA on a single A100 GPU with BF16 precision, as shown in Table 19. To support language feedback during evaluation (i.e., CoT inference), in the "Thinking" setting, we enable VLM auto-regressive generation every 20 action expert steps. The "Action Only" setting bypasses language generation and directly decodes latent actions via a single VLM forward pass. In the "Latent Action Caching", latent actions are generated every two expert steps; this introduces minimal performance impact. All settings are tested without action chunking. Note that although the model predicts 16-step action sequences, only one step is executed.

Table 19: **Inference speed.** Inference speed is evaluated under three settings **without using action chunking**. Each evaluation includes a 50-step warm-up followed by 200 steps for stable measurement.

|  | With Language | Action Only | Latent Action Caching |
|---|---|---|---|
| Inference Frequency(Hz) | 2.51 | 3.50 | 4.96 |

## F.6 EXPERIMENTS COMPUTE RESOURCES

The action pretraining phase requires approximately 27 hours on 64 A100 GPUs, with each node equipped with 1 TB of CPU memory. The VLA-IT phase takes about 12 hours under the same GPU configuration. Simulator-based evaluations are conducted with 8 A100 GPUs, while real-world experiments involve 4 hours of training on 32 A100 GPUs and deployment on a single A100 GPU.

To assess minimal training resources, we further reproduce pretraining results using 8 A800 GPUs in 2.5 days as shown in Table 20.

Table 20: **Evaluation results under different training settings.** We report mean success rates ($\%\pm$ standard error) across tasks, with Overall denoting the average over all tasks. "Main" corresponds to the results reported in the main table.

| Setting | Google Robot (VA) | | | | Google Robot (VM) | | | | WidowX Robot | | | | Overall |
|---|---|---|---|---|---|---|---|---|---|---|---|---|---|
|  | Pick Coke | Move Near | Drawer | Apple In | Pick Coke | Move Near | Drawer | Apple In | Put Spoon | Put Carrot | Stack Cube | Put Eggplant |  |
| Main (8 epochs) | 92.3±0.7 | 71.9±1.3 | 61.7±0.8 | 33.1±2.5 | 79.6±1.9 | 68.3±3.1 | 52.3±3.8 | 50.3±3.8 | 43.1±6.4 | 40.3±14.6 | 9.7±9.6 | 94.4±2.4 | 56.2±2.9 |
| 8 GPUs (4 epochs) | 94.0±0.2 | 76.9±0.5 | 62.8±1.6 | 39.3±4.3 | 88.7±1.7 | 67.4±2.1 | 61.8±2.5 | 31.7±1.9 | 62.5±11.0 | 48.6±2.4 | 8.3±4.2 | 95.8±4.1 | 61.5±1.3 |

# G    MULTIMODAL EXAMPLES

Figure 31 illustrates InstructVLA's multimodal and embodied commonsense reasoning across diverse scenarios. The model demonstrates accurate visual inference (e.g., recognizing a dog via reflection, identifying synthetic images), basic scene text recognition, and reliable grounding of objects and colors. In manipulation tasks, it interprets high-level goals, predicts appropriate next actions, and verifies task completion. These capabilities showcase its integration of perception, language, and manipulation, enabling effective performance in complex daily-life scenarios.

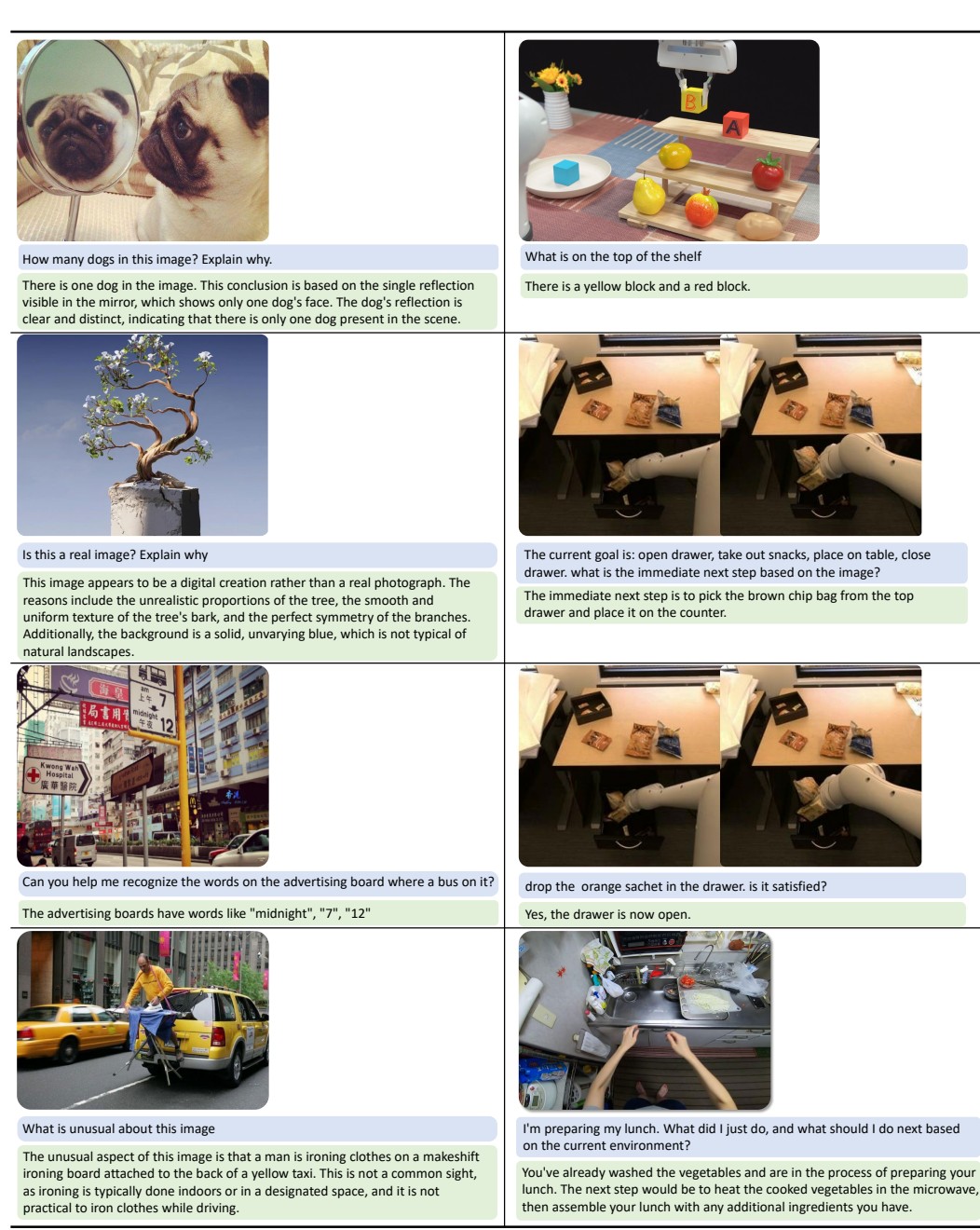

Figure 31: **Zero-shot multimodal question answering.** Four commonsense and four embodied examples are selected.

# H    REAL-WORLD EXPERIMENTS SETUP AND ANALYSIS

We collect data exclusively for few-shot settings as shown in Figure 32. In the first setting, which focuses on grasping objects in a clustered arrangement, the robot is instructed to classify objects within a $20 \times 40$ cm region on the table—placing all cubic objects into a plate and all others into a box. This setting includes 70 complete episodes, totaling 677 pick-and-place actions. In the second setting, which emphasizes spatial actions, the robot is instructed to randomly grasp three objects from the top of a rack and place them into a plate. We collect 60 complete episodes for this setting, comprising 180 pick-and-place actions. The experimental setups are depicted in Figure 36.

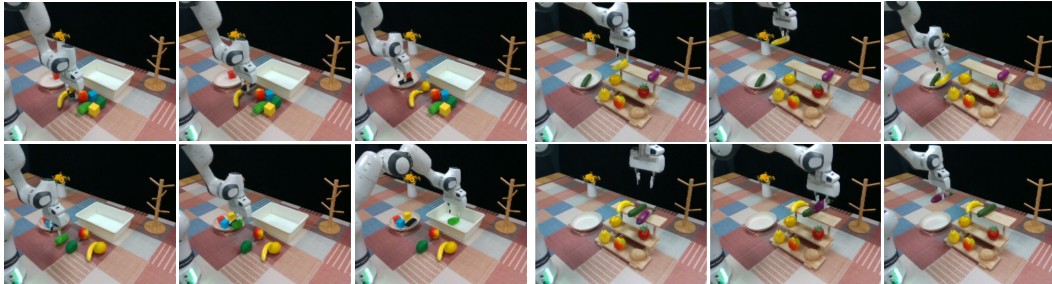

Figure 32: **Real-world dataset examples.** Four examples from the few-shot training set, illustrating cluster classification tasks (left) and rack pick-and-place tasks (right).

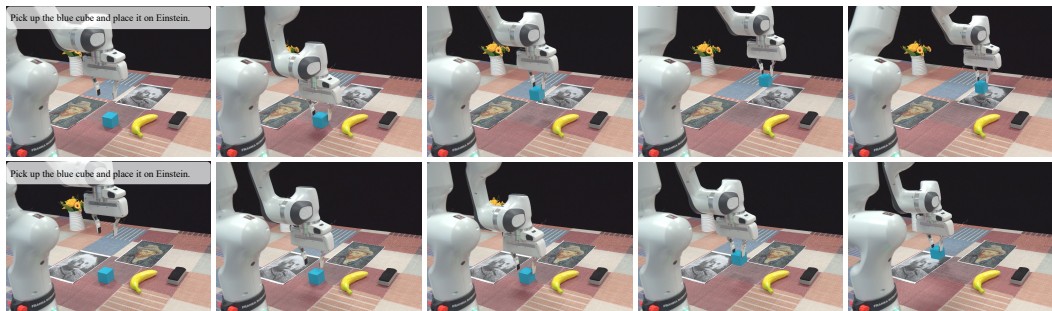

Figure 33: **Zero-shot grounding.** In a clustered pick-and-place setting, InstructVLA accurately places the blue cube by semantically grounding the reference to the celebrity.

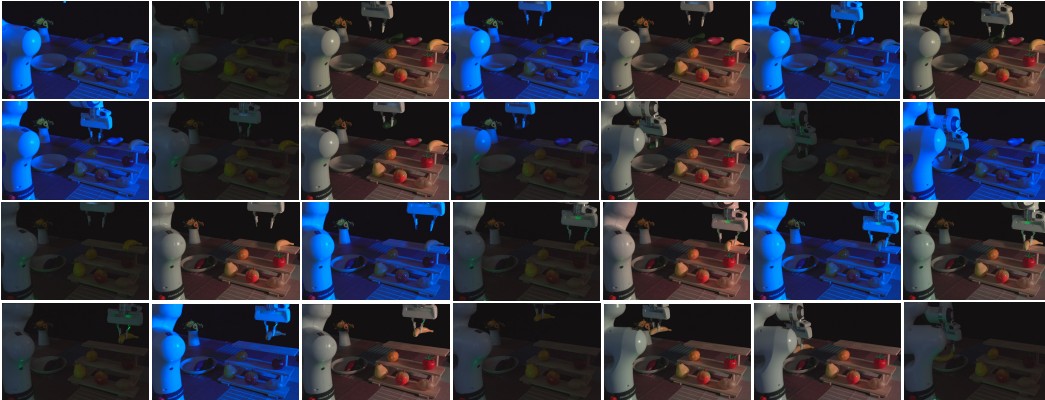

Figure 34: **Light distraction.** Stable visual features from DINO and SigLIP enable the model to operate robustly under extreme out-of-distribution lighting conditions.

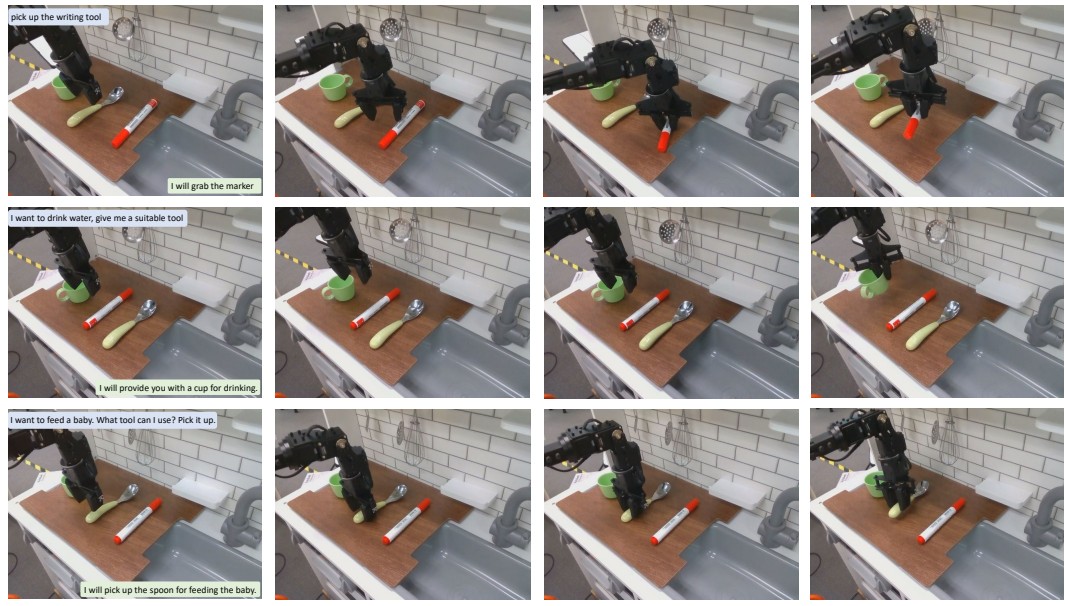

Figure 35: **Zero-shot evaluation.** We perform zero-shot evaluation in the Bridge kitchen environment with augmented background and novel objects. The instruction and model response are visualized in the first image.

To assess semantic grounding in novel contexts, we replace the plate and box in the cluster classification setting with images of celebrities. As illustrated in Figure 33, the model accurately interprets instructions and places the blue cube correctly by leveraging object and celebrity recognition.

Figure 34 shows that InstructVLA remains robust under extreme lighting conditions, supported by stable visual features from DINO and SigLIP. Finally, we evaluate zero-shot generalization in the Bridge kitchen environment with augmented backgrounds and unfamiliar objects. As shown in Figure 35, the model successfully follows novel instructions and completes the tasks.

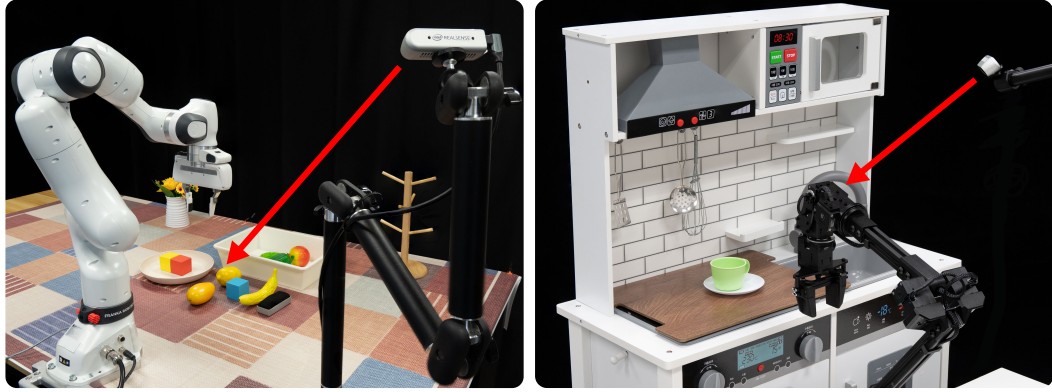

Figure 36: **Real-world settings.** A third-person view is captured using an Intel D435i camera for the Franka (few-shot) and WidowX (zero-shot) settings.

# I BROADER IMPACTS AND FUTURE WORK

## I.1 LIMITATION

InstructVLA integrates world knowledge into manipulation tasks by performing multimodal reasoning prior to action generation. Recent VLMs also excel at long-context processing and multi-turn dialogue. This motivates curating interleaved manipulation and reasoning with multi-turn interaction to support long-horizon tasks involving user intervention or reasoning-action alternation (Yao et al., 2023). Furthermore, the existing tasks are limited to basic primitives such as open/close and pick/place due to the constraints of the datasets we use (Brohan et al., 2022; Ebert et al., 2021) and the capabilities of the simulator. In contrast, standard VLM benchmarks typically contain thousands of tasks. Extending InstructVLA and SimplerEnv-Instruct bench to more dexterous skills is essential for real-world deployment.

## I.2 LLM USAGE STATEMENT

We employed large language models (LLMs) solely for grammar refinement and minor linguistic polishing. All LLM-assisted edits were carefully reviewed and verified by the authors to ensure that no fabricated content or unintended alterations to the original meaning were introduced. The research ideas, experimental design, data analysis, and conclusions presented in this work were entirely conceived and executed by the authors without LLM assistance.

## I.3 BROADER IMPACTS

InstructVLA contributes to the advancement of general-purpose embodied agents by integrating vision-language understanding with action generation. Its ability to follow free-form instructions and generalize to novel tasks supports applications in assistive robotics and human-robot collaboration. Nonetheless, as with other large pretrained models, careful attention must be given to potential limitations such as dataset bias and safety in real-world deployment. Ensuring responsible use and reliable performance across diverse environments is essential.

## I.4 FUTURE WORK

We plan to incorporate additional sensory modalities, such as depth and tactile feedback, to enhance safety and reliability in physical interactions. Leveraging recent advances in digital twins and simulation technologies, we aim to reduce reliance on real-world data by utilizing large-scale synthetic datasets. Finally, we will extend the evaluation and deployment of InstructVLA to a broader range of environments to further assess its generalization capabilities.

