# OpenReview forum: "Vision-Language-Action Instruction Tuning: From Understanding to Manipulation"
_ICLR.cc/2026/Conference — ICLR 2026 Poster_

### Official Review · Reviewer_kYXt · 2025-10-22

**Soundness:** 3
**Presentation:** 3
**Contribution:** 3
**Rating:** 6
**Confidence:** 4

**Summary:**

The authors argue that prior iterations of VLA models have not placed as much focus on embodied reasoning as they should. By embodied reasoning, we mean the ability of the model to move from being told exactly what manipulation to do, to a robot that can use context clues from the environment to decide what manipulation to do.

To address this, the authors introduce a new evaluation benchmark, based on augmenting the SimplerEnv setup with multilingual instructions, more indirect descriptions of objects (i.e. "I need something to clean with"), etc. They also train the VLA in a 2 stage process. First, an action expert is pretrained on a large manipulation dataset, to predict both learnable latent actions and language descriptions of the actions. This  is tuned against a fixed VLM backbone, using a flow matching loss. This approach was popularized by works like pi_0, but then as a 2nd stage, the authors train LoRAs on the LLM backbone against the fixed action expert. (Some experiments were done with finetuning the action expert too, and results were generally similar / slightly worse). The authors argue this 2nd stage can be viewed as a form of instruction tuning, as mediated by the action expert, to answer more complex questions.

This model is shown to perform reasonably on two prior benchmarks, a Google Robot and WidowX Robot. On the new SimplerEnv-Instruct benchmark, InstructVLA-Expert (the model trained only with the 1st stage) is only around on par with finetuned OpenVLA, but performs signficantly better after the stage 2 LLM backbone finetuning.

**Strengths:**

Paper does a good job of arguing that existing VLAs perform subpar at more complex queries, and the idea that the LLM backbone needs to be finetuned to exploit the action expert for embodied understanding seems reasonable. The proposed SimplerEnvInstruct benchmark is generally reasonable and gains on this benchmark are pretty significant. Ablations on different parts of the architecture, dataset, etc. are pretty extensive.

**Weaknesses:**

I think the paper is somewhat overreaching on how much it contributes to embodied reasoning. Even in the pre RT-1 days, there were some papers trying to do embodied reasoning. In my opinion the main difference these days is that the smaller, low latency VLMs are now more capable of answering these questions on their own.

Similarly, I find it difficult to attribute how much gain comes from the general evolved understanding that MoE backbones are better than dense backbones (many baseline models date to before this time).

Last, there is always some benchmark bias, where in general you expect a paper to do better on its self proposed benchmark compared to other models, the largest gains coming from the self-proposed benchmark gives me some pause.

nit: please label the y-axes in Figure 5, Figure 6. Please make it clearer how many trials were done for each eval number and/or include confidence intervals (why do these only exist in the SimplerEnv-Instruct columns?). In Table 1/2, I think including robot state is a significantly more important variable than the rest and would prefer if that were separated from the other results more in some way. (It is also unclear which baselines methods use robot state and which do not.)

**Questions:**

When creating SimplerEnv-Instruct, what variations were considered? I see that there are around 80 tasks + 1.1k trials which necessarily limits coverage for practicality reasons, but I am curious what things were decided to not be covered.

---

> ### Author Response · Authors · 2025-11-21
> **Rebuttal by Authors (1/2)**
>
> We thank you for the constructive feedback. We appreciate your positive assessment of our motivation, finetuning strategy, dataset/architecture ablations, and the importance of evaluating embodied reasoning under realistic instructions. We respond to your concerns below.
>
> ---
>
> ## `1.`**Contribution to Embodied Reasoning.**
>
> > I think the paper is somewhat overreaching on how much it contributes to embodied reasoning. Even in the pre RT-1 days, there were some papers trying to do embodied reasoning. In my opinion the main difference these days is that the smaller, low latency VLMs are now more capable of answering these questions on their own.
>
> We agree that embodied reasoning has a long history in robotics, but we outline two major differences as follows:
>
> 1. Prior work has attempted embodied reasoning, but **only modern lightweight VLMs make end-to-end reasoning-guided manipulation feasible**, without extra planners or low-level policy.
> 2. What is **missing in recent embodied reasoning work is a systematic evaluation** of whether these **reasoning abilities are actually preserved and transferred to manipulation,** which our method and benchmark directly address.
>
> **Below we highlights key differs:**
>
> ---
>
> | Method | General Multimodal Ability | Manipulation Ability | End-to-End | Open Source | Dataset | Benchmark |
> | - | - | - | - | - | - | - |
> | **SayCan (2022)** | **✗**  | ✓ | **✗** | ✓ | **✗**   | **✗** |
> | **PaLM-E (2023)** | ✓ | **✗** | **✗**      | **✗** | **✗**   | **✗** |
> | **RT-2 (2023)** | **✗** | ✓ | ✓  | **✗**  | **✗**   | **✗** |
> | **ECoT (2024)** | **✗** | ✓ | ✓ | ✓ | ✓ | **✗** |
> | **InstructVLA (Ours)** | **✓** | **✓** | **✓** | **✓** | **✓** | **✓** |
>
> **Summary:** InstructVLA is the only approach that is capable across all dimensions, combining preserved multimodal reasoning with a compact, end-to-end manipulation system.
>
> ## `2.` **Effect of MoE Beyond Backbone Improvements**
>
> > Similarly, I find it difficult to attribute how much gain comes from the general evolved understanding that MoE backbones are better than dense backbones (many baseline models date to before this time).
>
> To isolate MoE’s contribution, we conducted an ablation removing the MoE module while keeping all other components intact. As shown in Figure 6 (b) :
>
> | Method | Backbone      | SimplerEnv Avg. (Manipulation) | MM-Vet (Multimodal) |
> | - | - | - | - |
> | OpenVLA (Dense)     | Prismatic VLM |  39.0 | 9.1 |
> | InstructVLA (Dense) | Eagle-2  VLM  | 32.6  |  45.4 |
> | InstructVLA  | Eagle-2  VLM  | **49.7**   | **47.5**            |
>
> **Key Observations:**
>
> 1. **Dense adaptation cannot balance reasoning and manipulation: optimizing for one inevitably harms the other.** OpenVLA(Dense) collapses in reasoning (9.1) and InstructVLA(Dense) loses manipulation (32.6).
> 2. **Only MoE achieves *both* strong reasoning and strong manipulation.** The full MoE model achieves strong manipulation (49.7) and strong multimodal reasoning (47.5), indicating that expert separation is essential for stable reasoning-guided control.
>
> ## `3.` **Performance on Self-Proposed Benchmark**
>
> > Last, there is always some benchmark bias, where in general you expect a paper to do better on its self proposed benchmark compared to other models, the largest gains coming from the self-proposed benchmark gives me some pause.
>
> We acknowledge the reviewer’s concern, but **the observed performance gap is not a benchmark bias**, but a difference in how models handle *reasoning-intensive* manipulation tasks. In SimplerEnv-Instruct (`Appendix A.1.2`), **we additionally provide an zero-shot ablation** showing that tasks require interpreting OCR cues and grounding relational semantics—capabilities that current VLAs often struggle with.
>
> **All results below are obtained *without* using any data from our proposed dataset during training (Zero-shot), ensuring that the models are not biased toward these tasks.**
>
> | | Type of the Manipulation Tasks | OpenVLA | CogACT (OpenVLA+Head) | Magma | InstructVLA-Expert |
> | - | - | - | - | - | - |
> | Place L Near V  | Reasoning  | 16.7    | 8.33 | 33.3  | **58.3**  |
> | Place L Near A  | Reasoning | 18.8    | 0.0  |  25.0 | **50.0**  |
> | SimplerEnv Avg. | In-domain | 27.2    | 62.0  | 43.6  | 50.9  |
>
> **Key Observations:**
>
> - **CogACT** overfits to in-domain tasks (62.0 in-domain) but **collapses once reasoning is required**, dropping to **8.3 / 0.0**, **even worse than OpenVLA’s 16.7 / 18.8**.
>
> - **Magma** shows improved generalization (+16.6 / +6.2) with multimodal cotraining.
>
> - **InstructVLA** maintains multimodal reasoning and achieves the **strongest zero-shot performance** (**58.3 / 50.0**).
>
>
> **To further ensure fairness, we include:**
>
> - comparisons against multiple VLA models
> - ablations on OpenVLA under the same data recipes (Appendix A.4)
> - further extended to widely used agentic pipeline (fine-tuned OpenVLA + GPT-4o).
>
> **We hope this benchmark motivates future work on embodied reasoning.**

---

> > ### Author Response · Authors · 2025-12-01
> > **Rebuttal by Authors (2/2)**
> >
> > ## `4.` **Clarifications on Figures, Confidence Intervals, and Robot State**
> >
> > > nit: please label the y-axes in Figure 5, Figure 6. Please make it clearer how many trials were done for each eval number and/or include confidence intervals (why do these only exist in the SimplerEnv-Instruct columns?). In Table 1/2, I think including robot state is a significantly more important variable than the rest and would prefer if that were separated from the other results more in some way. (It is also unclear which baselines methods use robot state and which do not.)
> >
> > 1. **y-axis labels**
> >    Thank you for pointing this out. The y-axis represent the  success rate; we have corrected it.
> >
> > 2. **Number of trials and confidence intervals**
> >    We evaluated all benchmarks using three random seeds.
> >
> > - SimplerEnv: ~9k total trials (3k × 3 seeds)
> >
> > - SimplerEnv-Instruct: ~3k total trials (1k × 3 seeds)
> >
> >   We primarily report averages for simplicity. Confidence intervals are included for SimplerEnv-Instruct to demonstrate the robustness and stability of the proposed benchmark. **We further provide the confidence intervals on SimplerEnv**:
> >
> >   |      | InstructVLA Expert | InstructVLA Expert(s) | InstructVLA Generalist | InstructVLA Expert(s) |
> >   | ---- | ------------------ | --------------------- | ---------------------- | --------------------- |
> >   | Avg. | 50.9±3.6           | 61.2±2.2              | 49.7±0.9               | 54.9±0.5              |
> >
> > 3. **Robot state usage**
> >    We agree that the inclusion of robot state is an important factor. For each baseline, we use their official setting (w or w/o state). For fairness, we include our implementation also with and without state. **Both our state and no-state implementations outperform the baselines**.
> >
> >    For clarity, we explicitly list which baselines use robot state, and we mark them in the paper using the same notation (S.).
> >
> > | with state            | w/o state                                  |
> > | --------------------- | ------------------------------------------ |
> > | RoboVLMs, GR00T, Pi-0 | RT-1-X, RT-2-X, OpenVLA, SpatialVLA, Magma |
> >
> > ## `5.` **Task Coverage in SimplerEnv-Instruct**
> >
> > > When creating SimplerEnv-Instruct, what variations were considered? I see that there are around 80 tasks + 1.1k trials which necessarily limits coverage for practicality reasons, but I am curious what things were decided to not be covered.
> >
> > - **Benchmark structure.**
> >
> >   The benchmark follows a two-axis design: (1) **Task Aggregation**, which covers robustness and generalization variants aligned with prior VLA evaluations; and (2) **Situated Reasoning**, a newly introduced axis that targets scenarios where key information is implicit and must be inferred from environmental context. This reasoning axis is unique to our benchmark and is not supported by existing VLA datasets or protocols.
> >
> > - **Scope and exclusions.**
> >
> >   Tasks are curated to keep evaluation practical while still probing both generalization and reasoning. We exclude tasks unsupported by the underlying datasets (e.g., requiring high-precision deformable manipulation) and tasks whose success cannot be evaluated reliably in simulation (e.g., wiping, folding). These choices ensure the benchmark focuses on reasoning rather than being confounded by data limitations or unstable physics. We also note these exclusions in the limitations section.
> >
> > - **Scalability.**
> >
> >   Our benchmark and data generation pipeline are scalable, and we plan to incorporate large-scale datasets such as DROID and adopt more advanced simulators like IsaacSim to further expand task diversity and realism.

---

### Official Review · Reviewer_5dRE · 2025-10-27

**Soundness:** 2
**Presentation:** 3
**Contribution:** 3
**Rating:** 4
**Confidence:** 5

**Summary:**

This paper introduces InstructVLA , a model designed to solve "catastrophic forgetting" in Vision-Language-Action (VLA) models, a common problem where models lose their pre-trained multimodal reasoning abilities after being fine-tuned on manipulation tasks. The authors propose a novel training paradigm called "Vision-Language-Action Instruction Tuning (VLA-IT)" , which uses a Mixture-of-Experts (MoE) framework and a new 650K-sample dataset to jointly optimize both textual reasoning and action generation. As a result, InstructVLA successfully preserves its powerful language and reasoning capabilities while delivering state-of-the-art manipulation performance , significantly outperforming baselines like OpenVLA on the new SimplerEnv-Instruct benchmark.

**Strengths:**

1. VLA-IT Paradigm: It introduces a novel training paradigm called "Vision-Language-Action Instruction Tuning (VLA-IT)". This paradigm uses a Mixture-of-Experts (MoE) adaptation framework to jointly optimize textual reasoning and action generation, trained on standard VLM corpora and a new 650K-sample VLA-IT dataset.
2. Two-Stage Training: The training is divided into two stages: (1) Action Pretraining, which trains a VLM-driven "action expert" ; and (2) VLA Instruction Tuning, which freezes the action expert and fine-tunes only the VLM backbone via the MoE module to handle complex instructions and multimodal reasoning.
3. New Dataset and Benchmark: The paper contributes two significant resources: (a) a 650K VLA-IT dataset with diverse instructions, scene captions, and Q&A pairs ; and (b) a new benchmark, SimplerEnv-Instruct, featuring 80 tasks designed to evaluate generalization, closed-loop control, and high-level instruction understanding in VLAs.

**Weaknesses:**

1. This work is the combination of MoE and the fine-grained instruction finetuning. It seems so incremental. Especially, I can't see any advantages from the design. The biggest contribution is that the author gives the a 650K VLA-IT dataset with diverse instructions, scene captions, and Q&A pairs.
2. More related works containing the step by step reasoning ability should be added and discussed like CoT-VLA.
3. Authors should give more attention to the motivation. Particularly when the paper said their problem about ''how to train robotic manipulation skills without suffering "catastrophic forgetting" of the powerful multimodal reasoning capabilities inherent in pre-trained Vision-Language Models (VLMs)'', I don't know how instruction tuning can solve this kind of problems.

**Questions:**

See weakness.

---

> ### Author Response · Authors · 2025-11-21
> **Rebuttal by Authors (1/3)**
>
> We thank you for the careful assessment and for acknowledging the value of the VLA-IT paradigm, two-stage training structure, and the contributions of both the dataset and benchmark. We address your concerns below.
>
> ---
>
> ## `1. `**Clarifying the Contribution**
>
> Below, we provide a concise clarification addressing two core misunderstandings: (1) our contribution is **not** an incremental combination of MoE and instruction tuning but a **complete paradigm for reasoning-guided manipulation**, and (2) the 650K VLA-IT dataset is **not** the primary contribution **in isolation**, as prior VLA architectures cannot effectively exploit it without our proposed training framework.
>
> ---
>
> ### `1.1` **Paradigm contribution**
>
> >  This work is the combination of MoE and the fine-grained instruction finetuning. It seems so incremental. Especially, I can't see any advantages from the design.
>
> We clarify that our contribution is **not** the combination of MoE and instruction tuning, but a **complete training paradigm** that systematically addresses *reasoning-guided manipulation*, a capability largely absent in prior VLA research. **Our design does not originate from the two technical components highlighted by the reviewer.** Instead, both the architecture and training stages are derived from the ***fundamental challenge*** we aim to solve:
>
> **How can we acquire manipulation skills without eroding the VLM’s multimodal reasoning, and how can such reasoning, in turn, enhance manipulation?**
>
> This problem is non-trivial because existing VLA are not designed for high-level generalization or reasoning-guided manipulation. **To our knowledge, this work provides the first end-to-end pipeline for reasoning-guided manipulation in VLAs, converging the following key components**:
>
> - A reasoning-oriented VLA dataset.
> - A two-stage training procedure emphasizing reasoning-guided manipulation.
> - A dedicated benchmark (SimplerEnv-Instruct).
>
> In this framework, **the MoE** you mentioned function only **as simple implementation rather than central contributions**.
>
> To summarize, we respectfully ask the reviewer to consider **the significance and novelty of addressing the *overall problem* of reasoning-guided manipulation**, rather than evaluating the contribution based on individual techniques in isolation.
>
> ---
>
> ### `1.2` **The Dataset Is Insufficient Without the Proposed Training Paradigm**
>
> > The biggest contribution is that the author gives the a 650K VLA-IT dataset with diverse instructions, scene captions, and Q&A pairs.
>
> We argue that **the dataset is not the sole factor enabling reasoning-guided manipulation.** To validate this claim, **we conduct additional data ablation and scaling experiments**. Using OpenVLA as a representative baseline, we perform ablations on SimplerEnv-Instruct:
>
> |                        | Type         | OpenVLA | OpenVLA + 100% VLA-IT Data | InstructVLA | InstructVLA + 100% VLA-IT Data |
> | ---------------------- | ------------ | ------- | -------------------------- | ----------- | ------------------------------ |
> | Task Aggregation       | Manipulation | 14.8    | 30.5                       | 21.6        | 43.3                           |
> | Situated Reasoning     | Manipulation | 13.6    | 17.4                       | 12.9        | 48.8                           |
> | **Average**            | Manipulation | 14.2    | 24.0                       | 17.3        | 46.0                           |
> | **MM-Vet Performance** | Multimodal   | 0       | 10.4                       | 53.8        | 51.7                           |
>
> **Key Observations:**
>
> 1. **OpenVLA exhibits severe catastrophic forgetting**: its multimodal score remains low (MM-Vet **0 → 10.4**) even with 100% VLA-IT data, and manipulation improves only modestly (**14.2 → 24.0**).
> 2. **Direct finetuning is suboptimal**, as OpenVLA still underperforms on reasoning tasks (Situated Reasoning Tasks only **17.4** with full data).
> 3. **InstructVLA eliminates catastrophic forgetting and scales effectively**, achieving strong multimodal reasoning (MM-Vet **53.8 / 51.7**) and the best reasoning-guided manipulation performance (Average **17.3 → 46.0**).
>
> **To summarize,** **the dataset is not a standalone contribution**. Prior VLM-to-VLA paradigms suffer from severe catastrophic forgetting and therefore cannot effectively leverage the VLA-IT dataset. InstructVLA provides a more scalable architecture for reasoning-guided manipulation.

---

> ### Author Response · Authors · 2025-11-21
> **Rebuttal by Authors (2/3)**
>
> ## `2.` **Regarding CoT-VLA and Related Works**
>
> > More related works containing the step by step reasoning ability should be added and discussed like CoT-VLA.
>
> We have discussed related **step-by-step text-based reasoning VLAs**, including ECoT[1] and Magma[2], **in the introduction, related works, and the extended related works section (Appendix B.2)**. Our approach offers a unique advantage by enabling *reasoning-guided manipulation* supported by strong multimodal understanding and manipulation capabilities. In contrast, ECoT exhibits limited multimodal ability (`Table 1`), and Magma cannot perform multimodal reasoning in embodied environments (`Appendix A.2 Table 9`). **For CoT-VLA specifically:**
>
> - CoT-VLA‘s contribution focuses on utilizing goal image generation to improve manipulation performance.
> - It cannot perform visual–language reasoning using strong VLM pretraining.
> - Its use of video generation as auxiliary supervision is computationally heavy, requiring 11K A100 GPU hours with 96 A100s (pretraining). By comparison, our pretraining can be completed in 2.5 days on only 8 A100 GPUs (`Appendix F.6`).
>
> We provide the direct comparison on LIBERO for full clarity:
>
> | Method      | LIBERO-Spatial | LIBERO-Object | LIBERO-Goal | LIBERO-Long | Avg      |
> | ----------- | -------------- | ------------- | ----------- | ----------- | -------- |
> | CoT-VLA     | 87.5           | 91.6          | 87.6        | 69.0        | 81.1     |
> | InstructVLA | **97.3**       | **99.6**      | **96.5**    | **89.8**    | **95.8** |
>
> Overall, **InstructVLA surpasses CoT-VLA by a large margin across all LIBERO suites without extra video pretraining**, improving the average score from 81.1 to 95.8.
>
> Following your suggestion, **we have extended the detailed related works** and compared CoT-VLA in `Appendix B.2`.
>
> [1] Zawalski, Michał, et al. "Robotic control via embodied chain-of-thought reasoning." arXiv preprint arXiv:2407.08693 (2024).
>
> [2] Yang, Jianwei, et al. "Magma: A foundation model for multimodal ai agents." Proceedings of the Computer Vision and Pattern Recognition Conference. 2025.

---

> ### Author Response · Authors · 2025-11-28
> **Does our response address your concerns?**
>
> We further clarify the relation between VLA Instruction Tuning and catastrophic forgetting as follows.

---

> ### Author Response · Authors · 2025-12-01
> **Rebuttal by Authors (3/3)**
>
> ## `3.` **Relation between VLA-Instruction Tuning and Catastrophic Forgetting**
>
> > Authors should give more attention to the motivation. Particularly when the paper said their problem about ''how to train robotic manipulation skills without suffering "catastrophic forgetting" of the powerful multimodal reasoning capabilities inherent in pre-trained Vision-Language Models (VLMs)'', I don't know how instruction tuning can solve this kind of problems.
>
> A central question raised by reviewers concerns the connection between catastrophic forgetting and the effectiveness of VLA-Instruction Tuning. **In this section, we clarify this relationship** by explaining:
>
> 1. **How catastrophic forgetting is prevented in our framework.**
> 2. **Why such preservation is essential for enabling reasoning-guided manipulation.**
>
> ---
>
> ### `3.1` **How the Catastrophic Forgetting is Solved**
>
> We would like to clarify our motivation to avoid misunderstanding. **VLA Instruction Tuning is not designed to solve catastrophic forgetting**. Instead, **preventing catastrophic forgetting is the prerequisite** that enables VLA Instruction Tuning to produce a policy that benefit manipulation from multimodal reasoning.
>
> In other words: **we first eliminate forgetting, so that instruction tuning can operate on a stable reasoning foundation.**
>
> **To eliminate catastrophic forgetting, we decouple multimodal reasoning from action generation and reconnect them through latent action**, where an action LoRA expert learns manipulation behavior without interfering with the backbone and an MoE adaptation enables reasoning to further benefit manipulation with dynamic switching between abilities (`Appendix A.1.1`). **Together, these designs ensure that manipulation learning preserves pretrained multimodal reasoning and effectively leverages it during VLA-Instruction Tuning**.
>
> We track checkpoints during Stage-2 training and compare them against OpenVLA, which is fine-tuned using the same data recipe:
>
> ---
>
> | Benchmark      | Type         | Base VLM | Ours ~0.25 epoch (3k steps) | Ours 1 epoch (Reported) | OpenVLA (FT 20k steps) |
> | - | - | - | - | - | - |
> | Chart QA       | Multimodal   | 82.3     | 81.0  | 81.7 | 1.4  |
> | Text QA        | Multimodal   | 79.1 | 77.5  | 77.7 | 2.5  |
> | Doc QA         | Multimodal   | 88.0     | 86.1 | 85.8 | 29.2  |
> | MMB            | Multimodal   | 74.6     | 76.0  | 76.1  | 18.9  |
> | MMMU           | Multimodal   | 43.1     | 45.3  | 44.2 | 26.0 |
> | MMStar         | Multimodal   | 56.4     | 53.2 | 56.2 | 28.2 |
> | **MM**        | Average      | 70.6     | 69.9 | **70.3** | 17.7 |
> | Google Robot   | Manipulation | N/A      | 29.4  | 37.5 | 16.7 |
> | WidowX         | Manipulation | N/A      | 57.4 | 62.0 | 52.2 |
> | **SimplerEnv** | Average      | N/A      | 43.4 | **49.7** | 34.5 |
>
> **Key Observations:** Our method achieves **efficient convergence with no catastrophic forgetting**: the multimodal performance stays nearly unchanged (70.6 → 70.3), whereas manipulation increases from 43.4 to 49.7, outperforming OpenVLA by a wide margin.
>
> ---
>
> ### `3.2` **Vision-Language-Action Instruction Tuning Efficiently Leverages the Preserved Multimodal Ability**
>
> To validate that vision-language-action instruction tuning effectively leverages pretrained multimodal abilities, **we compare two representative training paradigms:**
>
>  (1) **Action pretraining followed by post-training** (OpenVLA)
>
>  (2) **Multimodal co-training** where QA and manipulation are optimized as separate tasks (Magma).
>
> |                    | Type         | Full Finetune (OpenVLA) | Full-Finetune + Post-training (OpenVLA) | Co-training (Magma) | VLA Instruction Tuning (InstructVLA) |
> | - | - | - | - | -| - |
> | Task Aggregation   | Manipulation | 14.8  | 30.5 | 26.2 | **43.6** |
> | Situated Reasoning | Manipulation | 13.6 | 17.4 | 21.4  | **48.8**   |
> | MM-Vet             | Multimodal   | 0.0 | 9.1 | 34.1 | **51.3**  |
>
> **Key Observations:**
>
> 1. **OpenVLA suffers from catastrophic forgetting**, leading to weak reasoning (MM-Vet **0.0 → 9.1**) and limited reasoning-guided manipulation gains (**13.6 → 17.4**).
> 2. **Magma demonstrates less forgetting**(MM-Vet **34.1**) but struggles to benefit manipulation with multimodal ability.
> 3. **InstructVLA achieves the strongest reasoning-guided manipulation**, reaching **43.6** on Task Aggregation and **48.8** on Situated Reasoning while also leading MM-Vet (**51.3**), showing it uniquely leverages preserved multimodal ability.
>
> **Summary.** Together, these results demonstrate that **preserving multimodal reasoning through our Stage-1 pretraining is essential**: only **when catastrophic forgetting is eliminated** can VLA-Instruction Tuning effectively leverage pretrained VLM capabilities **to produce strong reasoning-guided manipulation**. InstructVLA benefits from this synergy and achieves the highest performance across all evaluation settings.

---

### Official Review · Reviewer_wyF1 · 2025-10-30

**Soundness:** 3
**Presentation:** 2
**Contribution:** 3
**Rating:** 6
**Confidence:** 4

**Summary:**

This paper introduces InstructVLA, which is a VLA model that preserves large vision-language reasoning capabilities while achieving state-of-the-art manipulation performance. To achieve this, it introduces VLA-IT training paradigm with a 650K dataset. To evaluate generalization, this paper also proposes SimplerEnv-Instruct benchmark. InstructVLA achieves good performance on multimodal, Simpler, and SimplerEnv-Instruct benchmarks.

**Strengths:**

This paper shows a good example of how to effectively keep the capability of LLM/VLM in a VLA system, which is a very critical problem for current VLA models. It also demonstrates how to further enhance the reasoning ability of VLA. The proposed dataset and benchmark can also be a good reference for later works.

**Weaknesses:**

Please see the questions.

**Questions:**

**1.** The description of the architecture and training paradigm is quite confusing.

   **1.1** For example, they mention an action lora, a language lora, MoE adaptation, and a scalar head. However, it is hard to find details about them in Figure 2. Are the MoE and language lora only activated in stage 2?

   **1.2** I'm also confused about the training details of stage 1. The authors mention that they use the data of RT-1, and the model needs to predict both action and language, so what is the input of the model in stage 1?  If the task prompt and images are the input, is the output language generated by the authors?

   **1.3** In stage 1, what is the meaning of the action lora? Where exactly is this action lora applied? Do both language tokens and action tokens pass through the action lora? If only action tokens go through it, then what’s the point of the language loss? Is it just used as a simple regularizer?

**2**. The metric of the SimplerEnv-Instruct benchmark is success rate?
**3.** Table 2 shows that the training stage 2 is harmful to the normal simpler tasks, and there is little explanation about it. Since the necessity of reasoning VLA data is critical, can the authors provide more discussion?

---

> ### Author Response · Authors · 2025-11-21
> **Rebuttal by Authors (1/2)**
>
> We thank you for highlighting the importance of preserving VLM reasoning in VLA systems and for recognizing the relevance of our proposed dataset and benchmark. Below, we provide detailed clarification regarding the architectural and training procedures.
>
> ---
>
> ## `1.` **Clarifying the MoE Adaptation and Training Procedure**
>
> ### `1.1`  **Architecture: When Are the LoRA Experts and MoE Activated?**
>
>    > **1.1** For example, they mention an action lora, a language lora, MoE adaptation, and a scalar head. However, it is hard to find details about them in Figure 2. Are the MoE and language lora only activated in stage 2?
>
>    Yes. **Only the action LoRA is active in Stage-1**, while the **language LoRA and the scalar head (i.e., the MoE router) are introduced and activated only in Stage-2**(as stated in L236-237 & L242-243).
>
>    Stage-1 is dedicated to manipulation learning, so we adapt the VLM solely through the action LoRA, which specializes the backbone for action prediction.
>
>    Within Stage-2, the action LoRA adapter functions as one of the expert pathways in the MoE adaptation. As the model needs to acquire embodied reasoning, which in turn requires two additional components:
>
>    - **Language LoRA**: injects embodied reasoning ability into the VLM backbone and restores multimodal capacity.
>    - **Scalar Head (MoE router)**: computes a scalar weight that dynamically mixes the action LoRA and language LoRA, enabling the model to balance manipulation and reasoning demands.
>
>    Figure 2 provides a high-level schematic, which is why these module-level details are not shown. Following your suggestion, we now include the complete architecture and explanations of the action LoRA, language LoRA, and scalar head in `Appendix F.2 ; Fig. 30`.
>
> ### `1.2`  **Training Details of Stage-1**
>
>    > 1.2 The authors mention that they use the data of RT-1, and the model needs to predict both action and language, so what is the input of the model in stage 1? If the task prompt and images are the input, is the output language generated by the authors?
>
> Stage-1 receives **only images and the task prompt** as input.
>
> The model is trained to output:
>
> **(1) The continuous robot action** from RT-1/Bridge data.
>
> **(2) A short, primitive-level language description of that action (optional, depending on the prompt)**. This language description is **not provided in the dataset**; instead, **we annotate it automatically using a rule-based mapping** from low-level actions to discrete primitives (e.g., +x → “move right”,  -x → “move left”). A detailed description of the annotation rules and examples is provided in `Appendix D.1; Fig. 22`. We also add an extra `Tab. 16` to illustrate the data sources used in each stage.
>
> ### `1.3` **Role of the Action LoRA Adapter**
>
>    > In stage 1, what is the meaning of the action lora?  Where exactly is this action lora applied?
>
>    The action LoRA for action learning is applied directly to the VLM backbone, enabling the latent action (the feature connecting the VLM and the action expert) to better capture the VLM’s multimodal reasoning. Freezing the VLM backbone preserves pretrained knowledge but hurts manipulation performance, while LoRA provides a lightweight adaptation, as shown below:
>
>    |                           | Google Robot | WidowX Robot |
>    | ------------------------- | ------------ | ------------ |
>    | InstructVLA - action LoRA | 34.0         | 14.6         |
>    | InstructVLA               | 60.9         | 31.0         |
>
>    > Do both language tokens and action tokens pass through the action lora? If only action tokens go through it, then what’s the point of the language loss?
>
>    We do **not** hard-route specific tokens to specific adapters. As shown in `Appendix A.1.1` , all tokens pass through all adapters and contribute to the VLM’s generation process. The scalar head reweights their contributions dynamically, enabling the model to dynamically modulate action-oriented and language-oriented adaptations.
>
>    > Is it just used as a simple regularizer?
>
>    No, unlike a regularizer that constraints parameters or maintains performance, the stage-1's language loss provides explicit semantic supervision by requiring the model to predict a human-readable action primitive (e.g., “move right,” “lift,”). This leads to clearer and more stable action representations, resulting in a +9.2% gain.
>
> ## `2.` **Benchmark Metric**
>
>    > The metric of the SimplerEnv-Instruct benchmark is success rate?
>
>    Yes, the metric for SimplerEnv-Instruct in the main table is **success rate**, consistent with SimplerEnv.

---

> ### Author Response · Authors · 2025-11-21
> **Rebuttal by Authors (2/2)**
>
> ## `3.` **Why Stage-2 Affects Simpler Tasks**
>
> > Table 2 shows that the training stage 2 is harmful to the normal simpler tasks, and there is little explanation about it. Since the necessity of reasoning VLA data is critical, can the authors provide more discussion?
>
> Our results show that Stage-2 does **not** degrade performance on the original SimplerEnv tasks, while it delivers a **substantial boost** on the more challenging SimplerEnv-Instruct tasks that require stronger semantic and situated reasoning. To interpret this outcome, it is important to understand the distinction between the two benchmarks.
>
> First, note that original **SimplerEnv** focuses on low-level, less instruction-centric generalization across layout, objects, background. Thus, improvements in high-level reasoning are not expected to translate directly into gains on these simpler tasks.
>
> As stated in Sec. 4.2, **SimplerEnv-Instruct** captures two essential yet underexplored abilities in SimplerEnv: (1) policy generalization to linguistic and visual diversity, and (2) contextual reasoning in situated environments, evaluated in the situated reasoning suite. These are precisely the capabilities that VLA-IT is designed to strengthen.
>
> To ensure that our findings are not sensitive to a particular random seed, we performed an additional Stage-2 run with a different seed. This rerun verifies that SimplerEnv performance remains stable and that the gains on SimplerEnv-Instruct are highly consistent across seeds.
>
> | Model                    | SimplerEnv                      | SimplerEnv-Instruct                                          |                    |
> | ------------------------ | ------------------------------- | ------------------------------------------------------------ | ------------------ |
> |                          | Original, requires no reasoning | Requires unseen task generalization and reasoning (see task setups in Sec.4.2) |                    |
> |                          | Average                         | Task Aggregation                                             | Situated Reasoning |
> | Stage-1                  | 50.9                            | 21.6                                                         | 12.9               |
> | OpenVLA + GPT            | 39.0                            | 38.8                                                         | 32.4               |
> | Stage-2 (reported)       | 49.7                            | 43.6                                                         | 48.8               |
> | Stage-2 (different seed) | 51.6                        | 44.6                                                         | 49.4               |

---

> > ### Comment · Reviewer_wyF1 · 2025-11-27
> >
> > Thank you for your response. I will keep my positive score and raise my confidence (if a score of 7 were available, I would select it).

---

> > > ### Author Response · Authors · 2025-11-27
> > >
> > > Dear Reviewer wyF1,
> > >
> > > Thank you for your thoughtful and encouraging feedback. We’re glad our response strengthened your confidence in the submission. If you have any further questions or suggestions, please feel free to let us know.
> > >
> > > Best regards,
> > >
> > > Authors

---

### Official Review · Reviewer_vAo8 · 2025-11-04

**Soundness:** 4
**Presentation:** 4
**Contribution:** 3
**Rating:** 6
**Confidence:** 4

**Summary:**

The article proposes a novel VLA model architecture that uses MoE architecture to effectively address the critical challenge of catastrophic forgetting of pre-trained VLA capabilities. The work is exciting and solid, providing a clear narrative that successfully addresses the challenge of integrating multimodal reasoning ability into VLA models. The introduction of a dedicated evaluation task, SimplerEnv-Instruct, is also a substantial contribution to the field.

**Strengths:**

Effective Resolution of Core VLA Challenges and Strong Performance: The work successfully addresses major VLA obstacles, namely catastrophic forgetting of multimodal skills and the difficulty of tightly integrating high-level reasoning with precise low-level control. Also itsresults are compelling, validating the model’s efficacy.

Novelty in Training Paradigm and Architecture Design: The core technical strength lies in the Vision-Language-Action Instruction Tuning (VLA-IT) paradigm and its architectural implementation. This approach skillfully unifies autoregressive language generation with flow-based action generation using a Mixture-of-Experts (MoE) adaptation framework within a two-stage training strategy. This design allows for the dynamic alternation between textual reasoning and action execution , providing an efficient solution to task interference while preserving general VLM knowledge. and the diagram of this part is very clear.

The authors also curate a large-scale, customized 650K-sample VLA-IT dataset featuring detailed annotations for embodied scene understanding and planning, essential for bridging VLM knowledge with embodied scenes.

**Weaknesses:**

Lack of explaination of Architecture part. The proposed architecture, particularly the reliance on the Mixture-of-Experts (MoE) adaptation and the design of the latent action space, is central to the paper's claims of solving catastrophic forgetting and enabling unified reasoning-guided manipulation. However, key mechanistic details and the representational quality require further clarification to substantiate the claims.

Please provide a detailed description of the Scale Head (gating network) component of the MoE. Specifically, what is its architecture, what precise input does it take (e.g., is it the hidden state of the VLM?), and how is it trained during the Vision-Language-Action Instruction Tuning (VLA-IT) stage?

Also, the core challenge in co-training VLM capabilities and action skills is catastrophic interference. Did the training process, besides the standard $\mathcal{L}_{LM} + \mathcal{L}_{FM}$ loss3333, include any regularization loss (e.g., sparsity or orthogonality constraints) specifically designed to encourage or enforce the dynamic switching between the Language Adapter and the Action Adapter? Such detail is crucial to validate the claim that MoE effectively enables adaptation while preserving pre-trained knowledge.

**Questions:**

Please clarify the 31.7% increase in the performance part is a relative percentage or an absolute percentage increase. In my calculation its a relative percentage, but you sould clarify it for easier understanding.

---

> ### Author Response · Authors · 2025-11-21
> **Rebuttal by Authors**
>
> We sincerely thank you for the thoughtful and positive evaluation. We are encouraged by your recognition of (i) our solution to catastrophic forgetting in VLA models, (ii) the clarity of the VLA-IT training paradigm, and (iii) the contribution of the SimplerEnv-Instruct benchmark. We address your concerns below.
>
> ---
>
> ## `1.` **Request for Additional Architecture Explanation**
>
>    Following your suggestion, we added detailed MoE adaptation descriptions in `Appendix F.2 and Fig. 30` (scalar head architecture and its function). We further clarify that dynamic expert switching naturally arises without additional regularization, as follows.
>
> ### `1.1` **Clarification on Architecture: Scale Head (Gating Network)**
>
>    > Please provide a detailed description of the Scale Head (gating network) component of the MoE. Specifically, what is its architecture, what precise input does it take (e.g., is it the hidden state of the VLM?), and how is it trained during the Vision-Language-Action Instruction Tuning (VLA-IT) stage?
>
>    **(a) Architecture details**:  To maintain computational efficiency, the scalar head is designed as a lightweight 4-layer MLP applied on each token’s last hidden state from the VLM backbone. It takes the 2048-dimensional hidden states of the VLM as input, uses 128-dimensional intermediate layers, and outputs a 2-dimensional weight corresponding to the language and action LoRA adapters: 2048 $\rightarrow$ 128 $\rightarrow$ 128 $\rightarrow$ 128 $\rightarrow$  2.
>
>    **(b) Training details**: The scale head is optimized jointly with the LoRA adapters via gradients from both the language loss and action loss in VLA-IT stage. No additional regularization is introduced to keep the training procedure simple. Empirically, the pretrained action expert (stage-1) provides a strong inductive bias, enabling the router to prefer the appropriate expert depending on the task.
>
> ### `1.2`  **Dynamic Switching Without Dedicated Regularization Loss**
>
>    > Did the training process, besides the standard $\mathcal{L}*{LM} + \mathcal{L}*{FM}$ , include any regularization loss (e.g., sparsity or orthogonality constraints) specifically designed to encourage or enforce the dynamic switching between the Language Adapter and the Action Adapter ?
>
>    We do not apply explicit sparsity or orthogonality losses; instead, our method naturally encourages effective expert specialization through stage-1 pretraining.
>
>    During the stage-2(VLA-IT): the action expert is already optimized for manipulation signals (from stage-1 pretraining), while the language expert retains multimodal reasoning ability in the frozen backbone space. Routing more weight to the action expert exposes it primarily to action-loss gradients on manipulation tasks, whereas routing to the language expert exposes it to language-loss gradients on multimodal tasks. This natural gradient contrast implicitly encourages meaningful expert specialization. We track checkpoints during stage-2 training:
>
>    | Benchmark        | Base VLM | ~0.25 epoch (3k steps) | ~0.5 epoch (6k steps) | 1 epoch(Reported) |
>    | ---------------- | -------- | ---------------------- | --------------------- | ----------------- |
>    | chart QA         | 82.3     | 81.0                   | 81.3                  | 81.7              |
>    | Text QA          | 79.1     | 77.5                   | 77.3                  | 77.7              |
>    | Doc QA           | 88.0     | 86.1                   | 85.6                  | 85.8              |
>    | MMB              | 74.6     | 76.0                   | 77.4                  | 76.1              |
>    | MMMU             | 43.1     | 45.3                   | 43.8                  | 44.2              |
>    | MMStar           | 56.4     | 53.2                   | 56.7                  | 56.2              |
>    | **MM Average**   | 70.6     | 69.9                   | 70.4                  | 70.3              |
>    | **Manipulation** | N/A      | 43.4                   | 46.9                  | 49.7              |
>
>    Observations: Multimodal metrics remain stable, while manipulation performance steadily improves. In line with these results, we observe that the language loss converges within the first few training steps, whereas the action loss continues to decrease steadily. This indicates that router training is stable, dynamic switching emerges naturally, and no catastrophic forgetting occurs.
>
> ## `2.` **Clarification of “31.7% Improvement”**
>
>    > Please clarify the 31.7% increase in the performance part is a relative percentage or an absolute percentage increase. In my calculation its a relative percentage, but you sould clarify it for easier understanding.
>
>    Thank you for pointing this out. The reported “31.7% improvement” refers to a **relative** improvement. We have clarified this explicitly in the revision.

---

> ### Comment · Reviewer_vAo8 · 2025-11-24
>
> Thank you for the clarification. I will keep my score unchanged but increase my confidence. Good luck.

---

> > ### Author Response · Authors · 2025-11-27
> >
> > Dear Reviewer vAo8,
> >
> > We sincerely appreciate your time and effort in reviewing our work and providing insightful feedback. We’re glad our response has resolved your concerns. If you have any additional questions or suggestions, please feel free to let us know.
> >
> > Best regards, Authors

---

### Author Response · Authors · 2025-11-21
**General Response**

**Dear Reviewers, ACs, and SACs,**

We sincerely thank all reviewers for their thoughtful and constructive feedback. We appreciate the recognition of our motivation, benchmark, training methodology, and findings. Reviewers highlighted the importance of:

- **Addressing catastrophic forgetting in VLA models** *(vAo8, wyF1, 5dRE)*,
- **Preserving multimodal ability while demonstrating the transfer of multi-modal reasoning to manipulation** *(vAo8, wyF1, kYXt)*
- **The clarity of the two-stage training pipeline and  VLA-IT design** *(vAo8, 5dRE, kYXt)*,
- **The contribution of the 650K VLA-IT dataset and SimplerEnv-Instruct benchmark.** *(vAo8, wyF1, 5dRE, kYXt)*,

We are encouraged that reviewers found the problem well-motivated and the results compelling. We have incorporated these suggestions, and added new analyses to further strengthen the submission.

---

**Summary of Key Revisions and Additions**

- A full description of the MoE adaptation architecture `(Appendix F.2; Fig. 30)`. (vAo8, wyF1)
- Clearer Stage-1 language output annotation details and examples `(Appendix D.1; Fig. 22)`. (wyF1)
- New analyses showing how experts specialize without extra regularizations.  (vAo8, wyF1)
- Clarification of the motivation of InstructVLA. (5dRE)
- Expanded comparisons to CoT-VLA `(Appendix B.2)`. (5dRE )
- Clearer explanation of benchmark metrics, robot state settings and task design decisions. (kYXt)

---

All revisions are marked in `blue`. We appreciate the reviewers’ feedback and remain committed to improving the work. The following sections address each reviewer’s comments in detail.

---

> ### Author Response · Authors · 2025-12-01
> **A Summary of Our Rebuttal**
>
> `Reviewer vAo8`
>
> - **Main concerns:** `(1)` Clarification of MoE components (Scale Head, LoRA experts) and how dynamic routing is trained;  `(2)` Whether expert switching requires additional regularization;  `(3)` Clarification of ``31.7% improvement'' definition.
>
> - **Our response:** `(1)` A full architectural specification of the Scale Head, and detailed explanation of how LoRA experts are activated in Stage-2;  `(2)` **Empirical evidence** that dynamic routing emerges naturally without sparsity/orthogonality losses, supported by training-curve analysis and stability of multimodal metrics; `(3)` Explicit clarification that ``31.7%'' refers to **relative** improvement.
>
> - **Reviewer's Corresponding Comments: The concerns are resolved, and confidence increases(4→5).**
>
> ---
>
> `Reviewer wyF1`
>
> - ***Main concerns:***  `(1)` Missing details on when MoE and LoRA experts are activated;  `(2)` Stage-1 training inputs/outputs, and whether language descriptions exist in data; `(3)` Meaning and placement of the action-LoRA adapter; `(4)` Why Stage-2 affects SimplerEnv and whether reasoning data is necessary.
>
> - ***Our response:*** `(1)` A full timeline of module activation , now included in Appendix F.2 / Fig. 30; `(2)` Clarification that Stage-1 predicts continuous robot actions + automatically generated primitive-level language descriptions; `(3)` Explanation that action-LoRA is applied to the VLM backbone and improves manipulation performance significantly; `(4)` **Additional experiments** with different random seeds showing Stage-2 **does not harm** SimplerEnv while greatly improving reasoning-heavy SimplerEnv-Instruct.
>
> - ***Reviewer's Corresponding Comments:* The concerns are resolved (willing to raise to 7 if possible), and confidence increases(4→5).**
>
> ---
>
> `Reviewer 5dRE`
>
> - ***Main concerns:*** `(1)` Clarify the *actual* contribution - concern that work is ``just MoE + instruction tuning''; `(2)` Whether dataset alone is the main contribution; `(3)` Relation of VLA-IT to catastrophic forgetting; `(4)` Comparison with step-wise reasoning method(CoT-VLA);
>
> - ***Our response:***  `(1)` A clear articulation that the contribution is a **full paradigm for reasoning-guided manipulation**, not the combination of MoE and IT; `(2)` **Extensive ablations** showing that **dataset alone cannot eliminate catastrophic forgetting**, and only InstructVLA can effectively leverage VLA-IT data; `(3)` Clarification that VLA-IT does **not** solve catastrophic forgetting - Stage-1 does; VLA-IT benefits only after forgetting is removed; `(4)` Expanded related-work discussion, **direct LIBERO comparison with CoT-VLA**, showing large and consistent margins;
>
> ---
>
> `Reviewer kYXt`
>
> - ***Main concerns:*** `(1)` Whether our contribution to embodied reasoning is overstated; `(2)` How much performance gain comes from MoE vs. better backbones; `(3)` Whether gains on our benchmark reflect bias; `(4)` Clarifications on figures, statistical reporting, and robot-state usage; `(5)` Task coverage decisions in SimplerEnv-Instruct.
>
> - ***Our response:*** `(1)` **A clear comparison table** showing InstructVLA is the **only** method combining multimodal reasoning, manipulation, end-to-end control, open-source release, dataset, and benchmark; `(2)` **MoE ablation** showing dense models cannot jointly optimize manipulation + reasoning; `(3)` **Zero-shot evaluations demonstrating no benchmark bias** and strong reasoning generalization; `(4)` Corrected figures, added y-axis labels, provided confidence intervals and number of evaluation trials; `(5)` Clear explanation of benchmark design, exclusion criteria, scalability plan, and future expansions.
>
> ---
>
> We have revised the manuscript to incorporate updated related work and new experimental results. For the reviewer who did not respond, we respectfully ask the AC/SAC to consider our clarifications and assess the merits of our contributions accordingly.

---

### Meta-Review · Area_Chair_GRgr · 2026-01-06

**Summary:**

The initial reviews are leaning favourable across the reviewers (4666), praising the motivation and training paradigm. The rebuttal has resolved open questions, in particular Reviewer 5dRE's (4) concerns about incremental contributions and lack of novelty. Multiple reviewers noted after the rebuttal that they are even more confident about their recommendations.

I recommend accepting this manuscript.

**Reviewer Concerns:**

See above / below.

**Reviewer Scores:**

Reviewer vAo8: 6 unchanged

Reviewer wyF1: 6 unchanged

Reviewer 5dRE: 4 likely unchanged, though the rebuttal does appear strong

Reviewer kYXt: 6 unchanged

---

### Decision · Program_Chairs · 2026-01-26

Accept (Poster)